# Thalamocortical excitability modulation guides human perception under uncertainty

Julian Q. Kosciessa [1,2,3 ✉], Ulman Lindenberger [1,2] & Douglas D. Garrett [1,2 ✉]

Knowledge about the relevance of environmental features can guide stimulus processing. However, it remains unclear how processing is adjusted when feature relevance is uncertain. We hypothesized that (a) heightened uncertainty would shift cortical networks from a rhythmic, selective processing-oriented state toward an asynchronous ("excited") state that boosts sensitivity to all stimulus features, and that (b) the thalamus provides a subcortical nexus for such uncertainty-related shifts. Here, we had young adults attend to varying numbers of task-relevant features during EEG and fMRI acquisition to test these hypotheses. Behavioral modeling and electrophysiological signatures revealed that greater uncertainty lowered the rate of evidence accumulation for individual stimulus features, shifted the cortex from a rhythmic to an asynchronous/excited regime, and heightened neuromodulatory arousal. Crucially, this unified constellation of within-person effects was dominantly reflected in the uncertainty-driven upregulation of thalamic activity. We argue that neuromodulatory processes involving the thalamus play a central role in how the brain modulates neural excitability in the face of momentary uncertainty.

[1] Max Planck UCL Centre for Computational Psychiatry and Ageing Research, Berlin, Germany. [2] Center for Lifespan Psychology, Max Planck Institute for Human Development, Berlin, Germany. [3] Department of Psychology, Humboldt-Universität zu Berlin, Berlin, Germany. ✉email: kosciessa@mpib-berlin.mpg.de; garrett@mpib-berlin.mpg.de

Adaptive behavior requires dynamic adjustments to the perception of high-dimensional inputs. Prior knowledge about the momentary relevance of specific environmental features selectively enhances their processing while suppressing distractors[1], which can be implemented via gain modulation in sensory cortex (for reviews see refs. [2,3]). Crucially, however, a priori information regarding feature relevance is not always available, and how the brain flexibly adjusts the processing of complex inputs according to contextual uncertainty remains unclear[4] (Fig. 1a).

We hypothesize that such uncertainty-related processing adjustments involve a switch between different cortical states (Fig. 1b). Specifically, selective gain control has been associated with *phasic* (i.e., phase dependent) inhibition of task-irrelevant stimulus dimensions during cortical alpha (~8–15 Hz) rhythms[5]. Conceptually, such rhythmic modulations of feedforward excitability[6,7] provide temporal "windows of opportunity" for high-frequency gamma synchronization in sensory cortex[8] and increased sensory gain[9]. However, specifically increasing the fidelity of single stimulus dimensions is theoretically insufficient when uncertain environments require joint sensitivity to multiple stimulus features[10,11]. Alternatively, transient increases to the *tonic* excitation/inhibition (E/I) ratio in sensory cortex provide a principled mechanism for such elevated sensitivity to—and a more faithful processing of—high-dimensional stimuli[12]. In electrophysiological recordings, scale-free 1/f slopes are sensitive to differences in E/I ratio[13] and vary alongside sensory stimulation[14]. Relatedly, sample entropy provides an information-theoretic index of signal irregularity that is highly sensitive to scale-free content and may thus similarly track excitability[15]. However, whether contextual demands modulate scale-free activity and/or entropy is unknown. We hypothesize that heightened uncertainty shifts cortical states from a regime of rhythmic excitability modulations in the alpha band (associated with a modulation of gamma-band activity) towards tonic excitability increases (as indexed via increased scale-free irregularity and neural entropy; Fig. 1b).

Such "state switches" in cortical network excitability may be shaped by both neuromodulation and subcortical activity (Fig. 1c). Neuromodulation potently alters cortical states and sensory processing[3,16], and noradrenergic arousal in particular may permit high sensitivity to incoming stimuli[17]. Yet, non-invasive evidence is lacking for whether/how neuromodulation affects contextual adaptability. Moreover, despite early proposals for thalamic involvement in attentional control[18], studies have dominantly focused on cortical information flow (e.g., ref. [19]), at least in part due to technical difficulties in characterizing thalamic contributions. Crucially, the thalamus provides a nexus for the contextual modulation of cortical circuits[20], is a key component of neuromodulatory networks[17,21], and robustly modulates system excitability via rhythmic and aperiodic membrane fluctuations[22,23]. However, human evidence is absent for a central thalamic role in cortical state adjustments at the service of behavioral flexibility.

Here, we aimed at overcoming this lacuna by assessing the effects of contextual uncertainty during stimulus encoding on cortical excitability, neuromodulation, and thalamic activity in humans. We performed a multi-modal EEG-fMRI

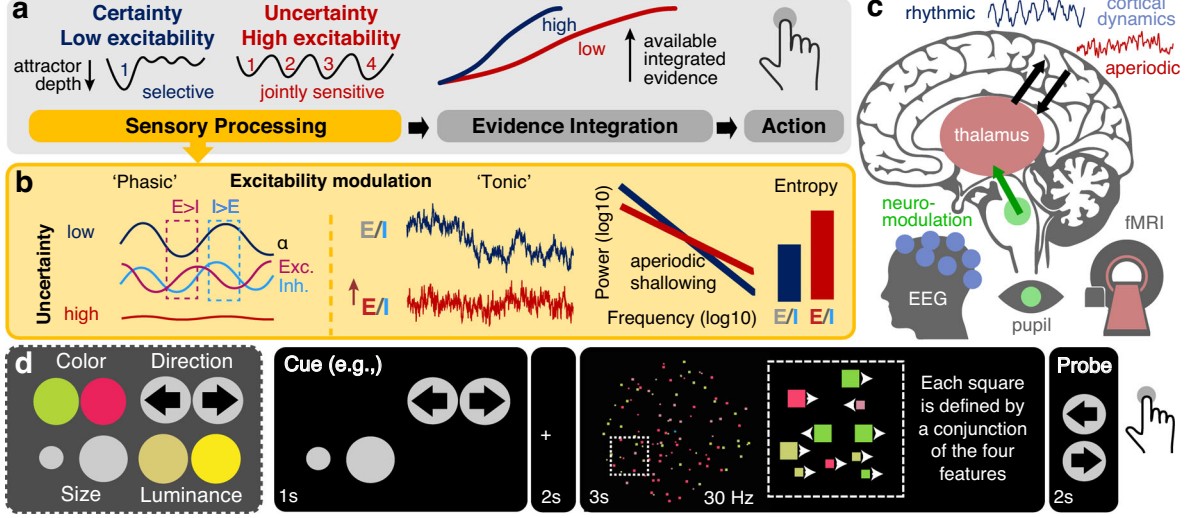

**Fig. 1 Hypotheses and task design. a** We probed whether uncertainty modulates cortical excitability during stimulus processing to guide subsequent evidence accumulation. We hypothesized that when cues specify the relevance of a single stimulus feature in advance, a low excitability regime may optimize subsequent choices via the targeted selection of relevant—and inhibition of irrelevant—information. In contrast, higher excitability may afford the concurrent sampling of multiple relevant features when the relevance of specific features is uncertain, but at the cost of a relative reduction of subsequently available evidence for any individual feature. **b** Hypothesized phasic and tonic excitability modulation in cortex as a function of uncertainty. Rhythmic (alpha) fluctuations in EEG are thought to reflect alternating phases of relative excitatory (E, Exc.) and inhibitory (I, Inh.) dominance[5,58,122], yielding variations in high-frequency (gamma) power as a function of low-frequency phase. In contrast, static increases in excitatory-to-inhibitory tone may alter aperiodic signal dynamics, reflected in a flattening of spectral slopes in the frequency domain[13] and relative increases in sample entropy as an index of signal irregularity. **c** We hypothesized that increasing probe uncertainty would induce a joint increase in neuromodulation and thalamic activity, associated with shifts from a phasic gain control mode (implemented via neural alpha rhythms) toward transient increases in excitatory tone (as indicated by aperiodic cortical activity). Participants participated in both an EEG and an fMRI session, allowing us to assess joint inter-individual differences in fast cortical dynamics (EEG) and subcortical sources (fMRI). **d** Participants performed a Multi-Attribute Attention Task ("MAAT") during which they had to sample up to four visual features in a joint display for immediate subsequent recall. On each trial, participants were first validly cued to a target probe set (here direction and size). The stimulus (which always contained all four features) was then presented for 3 s, and was followed by a probe of one of the cued features (here, which direction was most prevalent in the stimulus). The number and identity of simultaneous pre-stimulus cues were fixed for blocks of 8 trials to experimentally manipulate the level of expected (ongoing) probe uncertainty.

(electroencephalography-functional magnetic resonance imaging) experiment, measuring the same participants in two separate sessions, to capture both fast cortical dynamics (EEG) and sub-cortical activity (fMRI), while recording pupil dilation as a non-invasive proxy for neuromodulatory drive[24]. Participants performed a parametric adaptation of the classic dot motion task[25], in which individual stimulus elements were defined by a conjunction of color, size, direction and luminance features (Fig. 1d). By presenting valid cues (1–4 cues shown simultaneously) in advance of stimulus presentation, we manipulated the number of stimulus dimensions that are task-relevant in a given trial, while holding the sensory features of the task (i.e., its appearance on the screen) constant. Following the presentation of the multi-attribute stimuli, participants were probed with respect to a single-target feature that was selected from the cued set. By applying drift-diffusion modeling to participants' probe-related choice behavior while jointly assessing electrophysiological signatures of corresponding decision processes, we found that uncertainty during sensation reduced the rate of subsequent evidence integration, suggesting reductions in the precision of encoded target information. This uncertainty-related reduction in available evidence following the probe was associated with increased cortical excitability during the (pre-decisional) stimulus presentation period, as indexed by joint low-frequency (~alpha) desynchronization and high-frequency (~gamma) synchronization, and an increase in E/I ratio (as indicated by increased sample entropy and flatter scale-free 1/f slopes). These excitability adjustments, potentially reflecting a joint encoding of multiple features during periods of higher uncertainty, occurred in parallel with increases in pupil-based arousal. Finally, inter-individual differences in the modulation of cortical excitability, drift rates and arousal were jointly associated with the extent of thalamic blood oxygen level-dependent (BOLD) signal modulation, pointing to the importance of subcortical mechanisms for cortical state adjustments. Together, these findings suggest that neuromodulatory processes involving the thalamus shape cortical excitability states in humans, and that a shift from alpha-rhythmic to aperiodic neural dynamics adjusts the processing fidelity of external stimuli in service of upcoming decisions.

## Results

We developed a dynamic visual Multi-Attribute Attention Task ("MAAT") to uncover rapid adjustments to stimulus processing and perceptual decisions under expected uncertainty (Fig. 1d). Participants visually sampled a moving display of small squares, which were characterized by four stimulus features, with two exemplars each: their color (red/green), their movement direction (left/right), their size (large/small), and their color saturation (high/low). Any individual square was characterized by a conjunction of the four features, while one exemplar of each feature (e.g., green color) was most prevalent in the entire display. Multi-attribute stimuli were shown for a fixed duration of three seconds, after which participants were probed as to which of the two exemplars of a single feature was most prevalent (via 2-AFC). Probe uncertainty was parametrically manipulated using valid pre-stimulus cues, indicating the feature set from which a probe would be selected. The feature set remained constant for a sequence of eight trials to reduce set switching demands. Optimal performance required flexible sampling of the cued feature set, while jointly inhibiting uncued features; participants had to thus rapidly encode a varying number of targets ("target load") to prepare for an upcoming probe. Participants performed the task well above chance level for different features and for different levels of probe uncertainty (Supplementary Fig. 1a). As the number of relevant targets increased, participants systematically

became slower (median RT; EEG: $b = 0.14$, 95% CI $= [0.13, 0.15]$, $t(46) = 26.35$, $p = 2e-29$; MRI: $b = 0.11$, 95% CI $= [0.1, 0.12]$, $t(43) = 19.57$, $p = 5-23$) and less accurate (EEG: $b = -0.03$, 95% CI $= [-0.04, -0.03]$, $t(46) = -9.86$, $p = 6e-13$; MRI: $b = -0.02$, 95% CI $= [-0.03, -0.02]$, $t(43) = -7.5$, $p = 2e-9$) in their response to single-feature probes (Supplementary Fig. 1b).

**Probe uncertainty during sensation decreases the rate of subsequent evidence integration.** We leveraged the potential of sequential sampling models to disentangle separable decision processes in order to assess their modulation by probe uncertainty. In particular, drift-diffusion models estimate (a) the non-decision time (NDT), (b) the drift rate at which information becomes available, and (c) the internal evidence threshold or boundary separation (see Fig. 2a; for a review see ref. [26]). We fitted a hierarchical drift-diffusion model (HDDM) separately for each testing session and assessed individual parameter convergence with established EEG signatures. In particular, we investigated the centro-parietal positivity (CPP) and lateralized beta suppression as established neural signatures of evidence integration from eidetic memory traces[27]. The best behavioral fit was obtained by a model incorporating probe uncertainty-based variations in drift rate, NDT, and boundary separation (Supplementary Fig. 1c). Yet, there was no evidence for modulation of the threshold of the CPP or the contralateral beta response (Supplementary Fig. 1d). In line with prior work[28], we therefore selected an EEG-informed model with fixed thresholds across target load levels. With this model, reliability of individual parameters, as well as of their load-related changes, was high across EEG and MRI sessions (see below and Supplementary Fig. 1f, g). Parameter interrelations are reported in Supplementary Text 1.

Behavioral model estimates (Fig. 2b) and EEG signatures (Fig. 2c and Supplementary Fig. 2a) jointly indicated that probe uncertainty during stimulus presentation decreased the drift rate during subsequent evidence accumulation. This indicates a reduction of available evidence for single features when more features had to be sampled. Individual drift rate estimates for a single target were positively correlated with the slope of the CPP ($r = 0.52$, 95% CI $= [0.26, 0.71]$, $p = 3.59e-4$), while individual drift rate reductions reflected the shallowing of CPP slopes ($r(137) = 0.34$, 95% CI $= [0.18, 0.48]$, $p = 4.87e-5$). Notably, the magnitude of reductions in evidence accumulation with increasing probe uncertainty was strongly anticorrelated with the available evidence when the target attribute was known in advance (i.e., the single-target condition; EEG session: $r = -0.93$, 95% CI $= [-0.96, -0.88]$, $p = 2e-21$, MRI session: $r = -0.88$, 95% CI $= [-0.93, -0.78]$, $p = 2e-14$). That is, participants with more available evidence after attending to a single target showed larger drift rate decreases under increased probe uncertainty. Importantly, however, participants with higher drift rates for single targets also retained higher drift rates at higher probe uncertainty (i.e., high reliability for e.g., four targets: EEG: $r = 0.48$, 95% CI $= [0.22, 0.67]$, $p = 6e-4$; MRI: $r = 0.52$, 95% CI $= [0.25, 0.71]$, $p = 4e-4$). Moreover, individuals with higher drift rates across target loads exhibited lower average RTs (EEG: $r = -0.42$, 95% CI $= [-0.63, -0.16]$, $p = 0.003$; MRI: $r = -0.38$, 95% CI $= [-0.61, -0.08]$, $p = 0.01$) and higher task accuracy (EEG: $r = 0.86$, 95% CI $= [0.76, 0.92]$, $p = 2e-14$; MRI: $r = 0.89$, 95% CI $= [0.8, 0.94]$, $p = 3e-15$). Thus, in the present paradigm, more pronounced drift rate decreases with increasing probe uncertainty indexed a more successful modulation of feature-based attention during encoding and better overall performance.

We performed multiple control analyses to further elucidate decision properties. First, we did not observe a similar ramping of

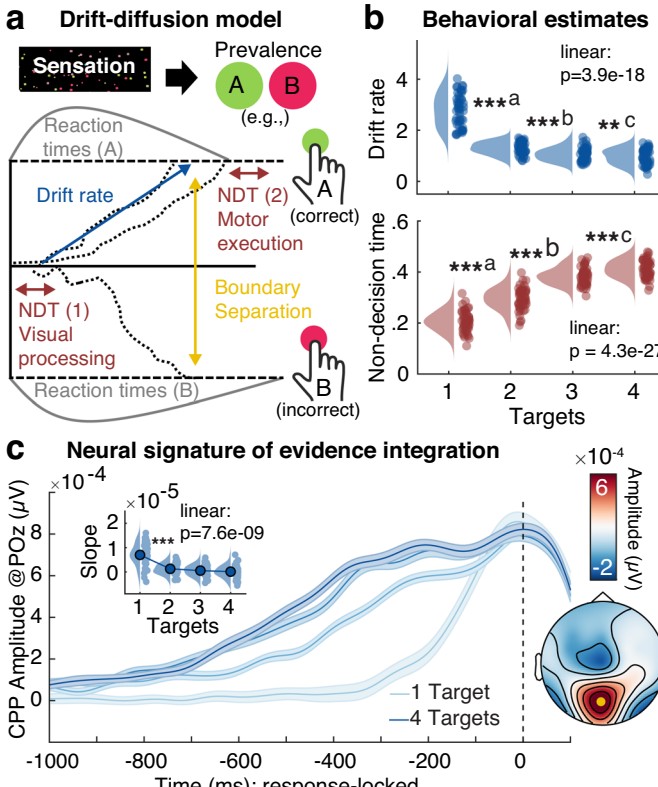

**Fig. 2 Evidence integration upon probe presentation decreases as a function of prior uncertainty. a** Schematic of drift-diffusion model. Following visual encoding, evidence is successively accumulated towards either of two bounds when probed for the dominant prevalence of one of two options of a single feature (e.g., color). A button press indicates the decision once one of the bounds has been reached and motor preparation has concluded. A non-decision time parameter captures visual encoding and motor preparation, drift rate captures the amount of available information, and boundary separation captures response bias (i.e., conservative vs. liberal). **b** Behavioral parameter estimates for drift rate and non-decision time (NDT; discussed in Supplementary Text 4), as indicated by the hierarchical drift-diffusion model (HDDM). Data are within-subject centered for visualization (see "Methods"; drift: ***a: $p = 2.9e{-}16$, ***b: $p = 6.5e{-}6$, **c: $p = 0.008$, linear: $b = -0.57$, 95% CI = $[-0.65, -0.49]$, $t(46) = -13.94$; NDT: ***a: $p = 5.6e{-}11$, ***b: $p = 8.2e{-}12$, ***c: $p = 0.0001$, linear: $b = 0.07$, 95% CI = $[0.06, 0.08]$, $t(46) = 23.27$). **c** Modulation of the centro-parietal positivity (CPP) as a neural signature of evidence accumulation (mean ± within-subject SEM). The response-locked CPP indicates decreases in pre-response integration rate with increasing probe uncertainty. Insets show CPP slope estimates from $-250$ to $-100$ ms relative to response execution (***$p = 7.2e{-}8$, linear: $b = -2e{-}6$, 95% CI = $[-2.8e{-}6, -1.5e{-}6]$, $t(46) = -7.05$), as well as the corresponding topography (CPP channel shown in yellow). Panels **b** and **c** indicate $p$ values from two-tailed paired $t$-tests, pairwise comparisons were Benjamini–Hochberg-adjusted for multiple comparisons. $n = 47$ participants for all panels. Source data are provided as a Source Data file.

the CPP during stimulus presentation (Supplementary Fig. 2b), suggesting that evidence accumulation was primarily initiated by the probe. Similarly, a decoding analysis of button responses indicated that information about choice execution was predominantly available following probe presentation, albeit with some pre-probe information when responses could be prepared in advance for single targets (Supplementary Text 2 and Supplementary Fig. 2c, d). Second, drift rate reductions were not primarily driven by differences between feature attributes (Supplementary Fig. 2e–g). Third, concurrent variations in

response convergence (i.e., trials in which the correct choices for all cued features converged on the same button response) could not account for the observed effects (Supplementary Text 3 and Supplementary Fig. 1e). Fourth, individual drift rates for single targets were unrelated to threshold estimates (EEG: $r = -0.05$, 95% CI = $[-0.33, 0.24]$, $p = 0.74$; MRI: $r = -0.06$; 95% CI = $[-0.35, 0.25]$, $p = 0.72$), thus suggesting a lack of differences in response bias[26]. Finally, participants with larger drift rate decreases exhibited more constrained NDT increases (EEG: $r(137) = 0.32$, 95% CI = $[0.16, 0.47]$, $p = 1.04e{-}4$; MRI: $r(122) = 0.37$, 95% CI = $[0.2, 0.51]$, $p = 2.48e{-}5$), indicating reduced additional motor transformation demands (see Supplementary Text 4) in high performers.

**Cortical excitability increases under uncertainty guide subsequent evidence integration.** Decreases in the rate of evidence integration indicate the detrimental consequences of probe uncertainty to single-feature decisions, but not the mechanisms by which sensory processing is altered. To investigate the latter, we examined rhythmic and aperiodic cortical signatures during stimulus processing. To jointly assess multivariate changes in spectral power as a function of probe uncertainty, we performed a partial-least-squares (PLS) analysis that produces low-dimensional, multivariate relations between brain-based data (in the present study, time–frequency–space matrices) and other variables of interest (see "Methods"). First, we assessed evoked changes compared to baseline using a task PLS, which here assesses optimal statistical relations between time-frequency matrices and experimental conditions. We observed a single latent variable (LV; permuted $p < 0.001$) that expressed jointly increased power in the delta–theta and gamma bands and decreased alpha power upon stimulus onset (Fig. 3a and Supplementary Fig. 4a), in line with increased cognitive control[29] and heightened bottom-up visual processing[8]. We next performed a task PLS to assess spectral power changes as a function of target load. A single LV (permuted $p < 0.001$; Fig. 3b–d) indicated a stronger expression of this control- and excitability-like pattern with increasing probe uncertainty. Next, we assessed the link between individual changes in multivariate loadings on this "spectral power modulation component" (SPMC) and behavioral modulations. We performed partial repeated measures correlations (see "Methods"), a mixed modeling approach that controls for the main effect of probe uncertainty in both variables of interest and indicates interindividual associations independent of the specific shape of condition modulation in individual participants. Crucially, participants with stronger spectral power modulation during sensation exhibited faster evidence integration in the single-target condition (Fig. 3f; 95% CI = $[0.27, 0.7]$), as well as stronger drift rate decreases under uncertainty ($r(137) = -0.4$, 95% CI = $[-0.53, -0.25]$, $p = 1.12e{-}6$; Fig. 3g), while showing constrained increases in NDT ($r(137) = -0.26$, 95% CI = $[-0.41, -0.1]$, $p = 0.002$). In sum, this suggests that high performers flexibly increased visual throughput as more features became relevant via top-down control of cortical excitability.

Here too, we performed multiple control analyses. First, the same multivariate power-band relations noted in our task PLS model (SPMC above) were also identified in a behavioral PLS model intended to estimate optimal statistical relations between power bands and behavior (Supplementary Text 5 and Supplementary Fig. 4b). [The main difference between task and behavioral PLS rests in the relation of multivariate neural values to categorical design variables in the former, and continuous individual "behavioral" variables in the latter[30].] Second, while we observed increases in pre-stimulus alpha power with increasing probe uncertainty, these changes did not relate to behavioral

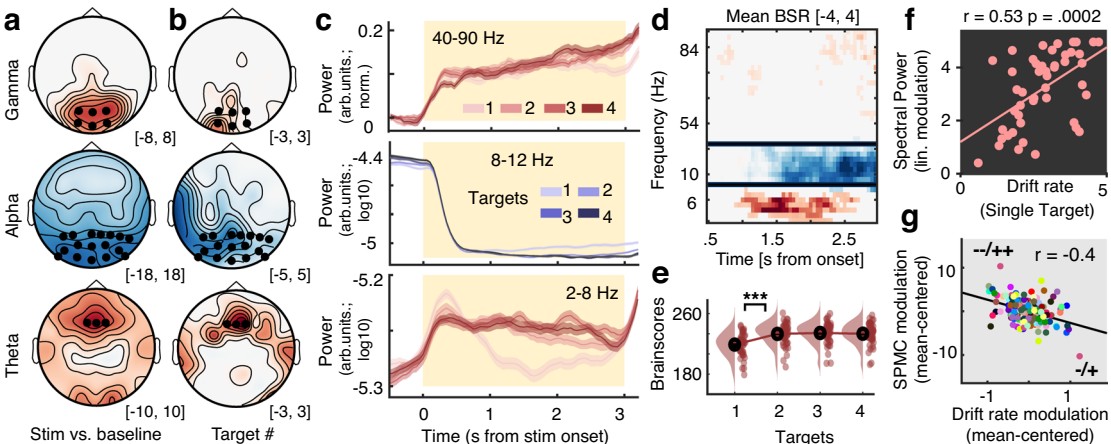

**Fig. 3 Multivariate power changes with probe uncertainty during stimulus encoding. a, b** Topographies of stimulus-evoked power changes relative to pre-stimulus baseline (**a**, see Supplementary Fig. 4a) and load-related power modulation (**b**). With increasing attentional demands, theta and "broadband" gamma power increased, whereas alpha rhythms desynchronized. Asterisks indicate the sensors across which data were averaged for presentation in **c** and **d**. Topographies indicate maximum (theta/gamma) or minimum (alpha range) bootstrap ratios (BSR) across time. **c** Temporal traces of band-limited power as a function of target load, extracted from the channels presented in **a/b** (mean ± within-subject SEM; $n = 47$ participants). **d, e** Multivariate loading pattern for **d** spectral power changes under uncertainty and **e** associated multivariate brain scores at different levels of target load ($n = 47$ participants; ***$p = 8.6e-20$; linear: $b = 3.33$, 95% CI = [2.82, 3.84], $t(46) = 13.11$, $p = 3.9e-17$). Black horizontal bars in panel **d** indicate discrete frequency ranges or sensors (shown in **a/b**). Data in **d** represent across-channel average BSR levels (each channel was first thresholded at a BSR of 3). These average BSRs thus provide a relative metric about how often the threshold was exceeded across channels, and every non-zero value in **d** indicates the presence of supra-threshold BSR values. Panel **e** indicates $p$ values from two-tailed paired $t$-test; pairwise comparisons were Benjamini–Hochberg-adjusted for multiple comparisons. **f, g** Participants with stronger multivariate power modulation exhibit stronger drift rates for single targets (**f**; $n = 47$ participants, Pearson's correlation) as well as stronger drift rate decreases under uncertainty (**g**). In **g**, dots represent linear model residuals (see "Methods"; $n = 188$: participants × target condition), colored by participant. Coupled changes across target conditions are indicated by the black line. We indicate the direction of main effects (first drift rate, then SPMC) via + and − (− = small decreases, −− = large decreases, + = small increases, ++ = large increases). SPMC spectral power modulation component. Source data are provided as a Source Data file.

changes or power changes during stimulus processing (Supplementary Text 6 and Supplementary Fig. 4c). Third, the entrained steady-state visual evoked potential (SSVEP) magnitude was not modulated by target load (Supplementary Text 7 and Supplementary Fig. 4d). Fourth, multivariate power changes corresponded to narrow-band, rhythm-specific indices in the theta and alpha band (Supplementary Text 8 and Supplementary Fig. 4e), and thus did not exclusively result from changes in the aperiodic background spectrum (see below).

**Alpha phase modulates gamma power during sensation.** Alpha rhythms have been related to phasic control over bottom-up input (presumably indexed by gamma power[8]). To assess phase-amplitude coupling (PAC) in the present data, we selected temporal alpha episodes at the single-trial level (see "Methods", Fig. 4a) and assessed the coupling between alpha phase and gamma power. We observed significant alpha–gamma PAC (Fig. 4b, d, left), consistent with alpha-phase-dependent excitability modulation. This was constrained to the occurrence of defined alpha episodes (see "Methods"), as no significant alpha-gamma PAC was observed prior to indicated alpha episodes (gray shading in Fig. 4a; Fig. 4d, right). Phasic gamma power modulation was observed across target load levels (Fig. 4f), but alpha duration decreased as a function of load (Fig. 4c). This suggests that alpha rhythms consistently regulated gamma power, but that alpha engagement decreased as more targets became relevant.

**Sample entropy and scale-free dynamics indicate shifts towards increased excitability.** Next, we assessed whether reduced alpha engagement was accompanied by increases in temporal irregularity, a candidate signature for system excitability[15,31]. We probed time-resolved fluctuations in sample entropy (SampEn), an

information-theoretic estimate of signal irregularity[32]. As sample entropy is jointly sensitive to broadband dynamics and narrow-band rhythms, we removed the alpha frequency range using band-stop-filters (8–15 Hz) to avoid contributions from alpha rhythms (see ref. [15]). A cluster-based permutation test indicated SampEn increases under probe uncertainty over posterior-occipital channels (Fig. 5a–c). Notably, the magnitude of individual entropy modulation in this cluster scaled with increases in the SPMC ($r(137) = 0.22$, 95% CI = [0.05, 0.37], $p = 0.01$). This indicates that multivariate changes in spectral power, including alpha desynchronization, were accompanied by broadband changes in signal irregularity.

Aperiodic, scale-free spectral slopes are a major contributor to broadband SampEn, due to their joint sensitivity to autocorrelative structure[15], and a shallowing of aperiodic ($1/f$) slopes has theoretically been associated with system excitability[13]. We therefore assessed aperiodic slope changes during the stimulus period (excluding onset transients). In line with our hypothesis, participants' PSD slopes shallowed under uncertainty (Fig. 5d–f), suggesting that participants increased their excitatory tone in posterior cortex. In line with the expectation that sample entropy should be highly sensitive to scale-free dynamics, sample entropy was strongly related to individual PSD slopes across conditions ($r = 0.78$, 95% CI = [0.64, 0.87], $p = 7e-11$) and to linear changes in PSD slope with increasing uncertainty ($r(137) = 0.44$, 95% CI = [0.3, 0.57], $p = 4.92e-8$). In sum, heightened probe uncertainty desynchronized low-frequency alpha rhythms and elevated the irregularity of cortical dynamics, in line with enhanced tonic excitability.

**Increases in phasic pupil diameter relate to transient spectral power changes.** Phasic arousal changes modulate perception and local cortical excitability (for reviews see refs. [3,16]). To test

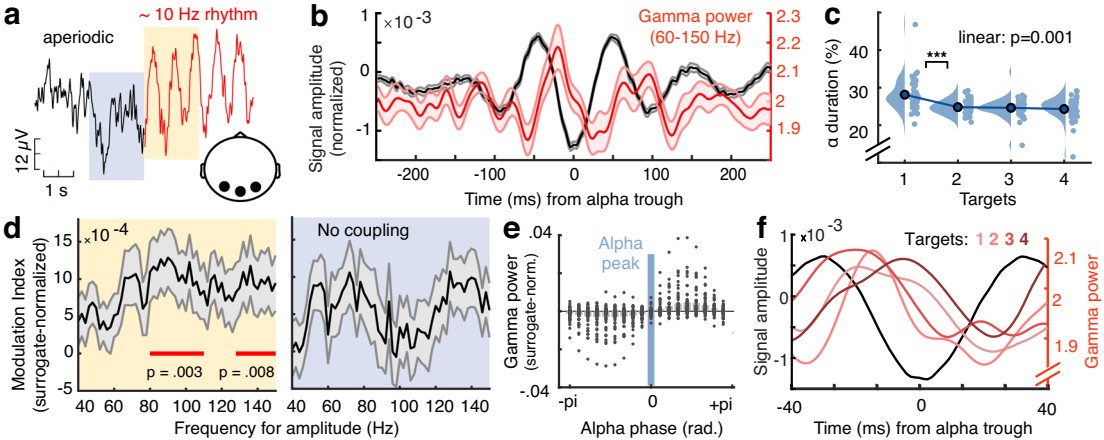

**Fig. 4 Alpha phase modulates gamma power during sensation. a** Exemplary time series around the onset of a detected alpha episode (example from four-target condition). Segments were pooled across occipital channels (black dots in inset topography) and target load conditions. **b** Normalized gamma power (red) during alpha episodes (yellow shading in **a**) is modulated by the alpha phase (see "Methods"). The unfiltered ERP aligned to the alpha trough is shown in black. Data are presented as mean ± SEM. **c** The relative duration of alpha (α) episodes decreased with increased feature relevance. Data are within-subject centered (see "Methods"). Panel **c** indicates p values from two-tailed paired t-tests (***p = 3.6e−5, linear: b = −1.17, 95% CI = [−1.81, −0.53], t(46) = −3.67), pairwise comparisons were Benjamini–Hochberg-adjusted for multiple comparisons. **d** Modulation index (MI) indicated significant coupling between the phase of alpha and gamma power during rhythmic events (left), but not during periods immediately prior to rhythm onset (right). MI was normalized using surrogate data to reduce erroneous coupling (see "Methods"). Panel **d** indicates p values from paired t-tests against zero, corrected for multiple comparisons via cluster-based permutations. Data are presented as mean ± SEM. **e** Gamma power (averaged from 60 to 150 Hz; dots indicate n = 47 participants) was maximal following alpha peaks. Power was normalized across all phase bins (see "Methods"). **f** Gamma power systematically peaks between the peak and trough of alpha rhythms across target levels. For this analysis, alpha episodes were collapsed across all participants. n = 47 participants in all panels. Source data are provided as a Source Data file.

whether arousal increased alongside uncertainty, we assessed phasic changes in pupillometric responses as a proxy for arousal during stimulus presentation. We quantified phasic pupil responses via the first temporal derivative (i.e., rate of change), as this measure has higher temporal precision and has been more strongly associated with noradrenergic responses than the overall pupil response[24]. Phasic pupil dilation systematically increased with probe uncertainty (Fig. 6). This modulation occurred on top of a general pupil constriction due to stimulus-evoked changes in luminance (Fig. 6a, inset), while the linear modulation occurred—by stimulus design—in the absence of systematic luminance differences across load levels.

Next, we assessed the relation between individual modulations in pupil diameter, cortical excitability, and behavior. The magnitude of pupil increases tracked increases in the spectral power modulation component (SPMC; r(137) = 0.22, 95% CI = [0.06, 0.38], p = 0.01), but did not directly relate to entropy (r(137) = −0.06, 95% CI = [−0.23, 0.1], p = 0.45) or aperiodic slope changes (r(137) = −0.04, 95% CI = [−0.2, 0.13], p = 0.67). Participants with larger increases in pupil dilation also exhibited higher drift rates at baseline (r = 0.31, p = 0.033), greater decreases in drift rate (r(137) = −0.17, 95% CI = [−0.33, 0], p = 0.05) and more constrained NDT increases (r(137) = −0.21, 95% CI = [−0.36, −0.04], p = 0.01) with increasing probe uncertainty. This suggests that arousal jointly related to spectral power changes during stimulus presentation and subsequent choices made at probe.

**Thalamic BOLD modulation tracks excitability increases during sensation.** Finally, we probed whether the thalamus acts as a subcortical nexus for sensory excitability adjustments under probe uncertainty. To allow spatially resolved insights into thalamic involvement, participants took part in a second, fMRI-based testing session during which they performed the same task. First, we investigated uncertainty-related changes in BOLD magnitude during stimulus processing via a task PLS targeting

optimal statistical differentiation of target load levels. This analysis suggested two reliable latent variables (LV1: permuted p ~ 0; LV2: permuted p = 0.007; Fig. 7). See Supplementary Table 1 for peak coordinates/statistics and Supplementary Fig. 5a/b for complete multivariate spatial patterns for the two LVs, with the first LV explaining the dominant amount of variance (89.6% crossblock covariance) compared to the second LV (8.7% crossblock covariance).

The first latent variable (LV1) indicated load-related increases dominantly in cortical areas encompassing the fronto-parietal and the salience network, as well as thalamus. Primary positive contributors to LV1 (i.e., representing increases in BOLD with increasing probe uncertainty) were located in mid-cingulate cortex (MCG), inferior parietal lobule (IPL), bilateral anterior insula (aINS), inferior occipital gyrus (IOG), thalamus, and bilateral inferior frontal gyrus (IFG). In contrast, relative uncertainty-related decreases in BOLD magnitude were dominantly observed in pallidum (potentially reflecting reduced motor preparation), bilateral posterior insula (pINS), left SFG, and left mid-cingulate cortex. Individual brain score increases were associated with stronger drift rate decreases (r(122) = −0.36, 95% CI [−0.5, −0.19], p = 5.11e−5), but not NDT, SPMC, or entropy (all p > 0.05). See Supplementary Text 9 for results from the second latent variable (LV2), which might reflect decreased engagement at higher levels of target uncertainty.

Finally, we performed a behavioral PLS examining optimal multivariate relations to various neuro-behavioral indices, to probe whether regional BOLD modulation tracked a unified set of individual differences in the modulation of cortical excitability, arousal, and behavior. We observed a single significant LV (permuted p = 0.001, 46.2% crossblock covariance) that dominantly loaded on anterior and midline thalamic nuclei with fronto-parietal projection zones (Fig. 7d), and extended broadly across almost the entirety of thalamus. BOLD magnitude increases were more pronounced in participants exhibiting higher drift rates (i.e., more available evidence) (r = 0.75, 95%

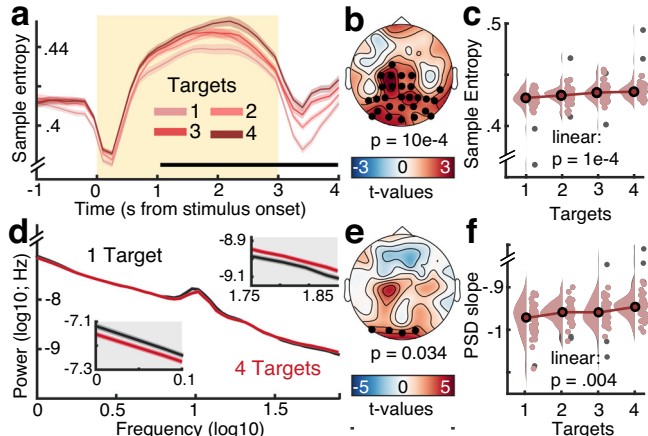

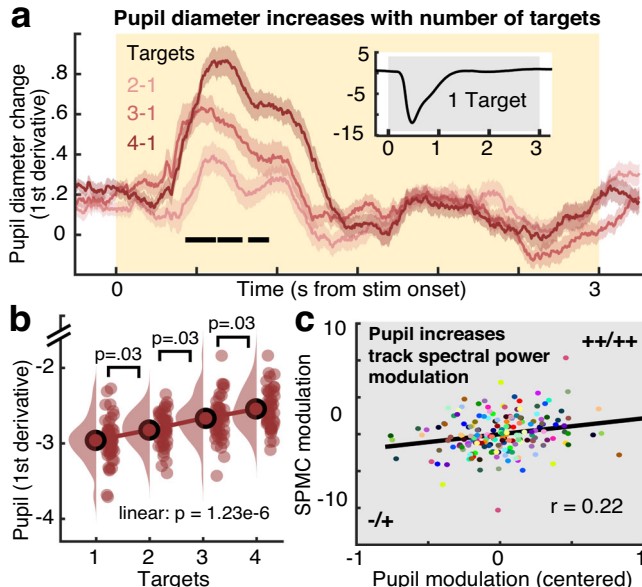

**Fig. 5 Uncertainty increases aperiodic dynamics during sensation.**
Aperiodic dynamics are estimated via neural sample entropy (**a–c**, SampEn) and 1/f power spectral density (PSD) slopes (**d–f**). **a** Temporal traces of sample entropy (mean ± within-subject SEM; $n = 47$ participants). The yellow background indicates the period of stimulus presentation. The black line indicates time points at which permutation tests indicated linear load effects.
**b** Topography of mean linear load effect estimates ($n = 47$), with black dots representing the significant cluster. **c** Post hoc analysis of entropy estimates within significant cluster. Gray dots indicate individual outliers (defined as Cook's distance > 2.5 × mean (Cook's distance); $n = 2$ participants) and have been removed from the statistical post hoc assessment ($n = 45$ participants). Data are within-subject centered for visualization (see "Methods"; linear: $b = 0.002$, 95% CI = [0.001, 0.003], $t(44) = 4.22$). **d** Aperiodic slopes shallow with increasing target load (i.e., spectral rotation across low- and high-frequencies; mean ± within-subject SEM; $n = 47$ participants). Lower and upper insets highlight slope differences at low and high frequencies, respectively. **e** Topography of mean linear load effects on 1/f slopes ($n = 47$). Black dots indicate the significant occipital cluster used for post hoc assessments. **f** Same as **c**, but for occipital aperiodic slopes. Data are within-subject centered for visualization (see "Methods"; $n = 3$ outliers in gray; effective $n = 44$ participants in red; linear: $b = 0.008$, 95% CI = [0.003, 0.012], $t(43) = 3.06$). Panels **b** and **e** indicate $p$ values from dependent samples regressions on target load, corrected for multiple comparisons via cluster-based permutations. Panels **c** and **f** indicate $p$ values from two-tailed paired $t$-tests (linear effects vs. zero). Source data are provided as a Source Data file.

**Fig. 6 Effect of probe uncertainty on pupil diameter as a proxy for neuromodulation. a** Phasic changes in pupil diameter increase with number of targets (mean ± within-subject SEM; $n = 47$ participants). Significant linear load effects as indicated by dependent samples regression with cluster-based permutation are indicated via the black line. For follow-up analyses, we extracted median pupil values from 0 to 1.5 s. For display purposes but not statistics, derivative estimates were smoothed via application of a 200 ms median running average. **b** Post hoc analysis of load effects in extracted median values. Data are within-subject centered for visualization (see "Methods"; $n = 47$ participants; linear: $b = 0.14$, 95% CI = [0.09, 0.19], $t(46) = 5.58$). Panel **b** indicates $p$ values from two-tailed paired $t$-tests; pairwise comparisons were Benjamini–Hochberg adjusted for multiple comparisons. **c** Coupled changes between our spectral power modulation component (SPMC) and pupil modulation. Dots represent linear model residuals (see "Methods"; $n = 188$: participants × target condition), colored by participant. We indicate the direction of main effects (first pupil, then SPMC) via + and − (− = small decreases, −− = large decreases, + = small increases, ++ = large increases). Source data are provided as a Source Data file.

bootstrapped (bs)CI = [0.72, 0.86]) and stronger drift reductions under probe uncertainty ($r = -0.6$, 95% bsCI = [−0.78, −0.54]; Fig. 7b), as well as lower baseline NDTs ($r = -0.37$, 95% bsCI = [−0.58, −0.08]), confirming that increased thalamic responses reflected behaviorally adaptive contextual adjustments. This association was specific to the behavioral adjustments of interest, as we noted no relations with NDT modulation ($r = 0.05$, 95% bsCI = [−0.31, 0.3]) or boundary separation ($r = 0.08$, 95% bsCI = [−0.24, 0.37]). Importantly, higher (dominantly thalamic) BOLD modulation was further associated with greater increases on the SPMC ($r = 0.31$, 95% bsCI = [0.16, 0.58]), in phasic pupil dilation ($r = 0.67$, 95% bsCI = [0.51, 0.81]) and in entropy assessed during the EEG session ($r = 0.22$, 95% bsCI = [0.08, 0.46]; Fig. 7b). 1/f shallowing was not stably related to BOLD modulation ($r = -0.24$, 95% bsCI = [−0.38, 0.19]), potentially due to noisier individual estimates. BOLD modulation was unrelated to chronological age ($r = -0.20$, 95% CI = [−0.14, 0.45], $p = 0.21$), gender (male vs. female; $r = -0.28$, 95% CI = [−0.54, 0.03], $p = 0.08$), subjective task difficulty (rated on 5-point Likert scale; $r = -0.02$, 95% CI = [−0.32, 0.28], $p = 0.89$), or framewise displacement of BOLD signals (an estimate of in-scanner motion; $r = -0.24$, 95% CI = [−0.51, 0.07], $p = 0.13$).

Taken together, these results suggest a major role of the thalamus in integrating phasic neuromodulation to regulate rhythmic and aperiodic cortical excitability according to contextual demands.

## Discussion

To efficiently process information, cortical networks must be flexibly tuned to environmental demands. Invasive studies indicate a crucial role of the thalamus in such adaptations (for a review see ref. [20]), but human evidence on thalamic involvement in rapid cortical regime switches at the service of behavioral flexibility has been missing. By combining a multi-modal experimental design with a close look at individual differences, we found that processing under contextual uncertainty was associated with a triad characterized by thalamic BOLD modulation, EEG-based cortical excitability, and pupil-based indicators of arousal. In light of this triad, we propose that thalamic regulation of sensory excitability is crucial for adaptive sensory filtering in information-rich environments.

By cueing relevant dimensions of otherwise physically identical stimuli, we observed that increases in the number of attentional targets reliably reduced participants' available evidence (as evidenced by drift rate decreases) during subsequent perceptual

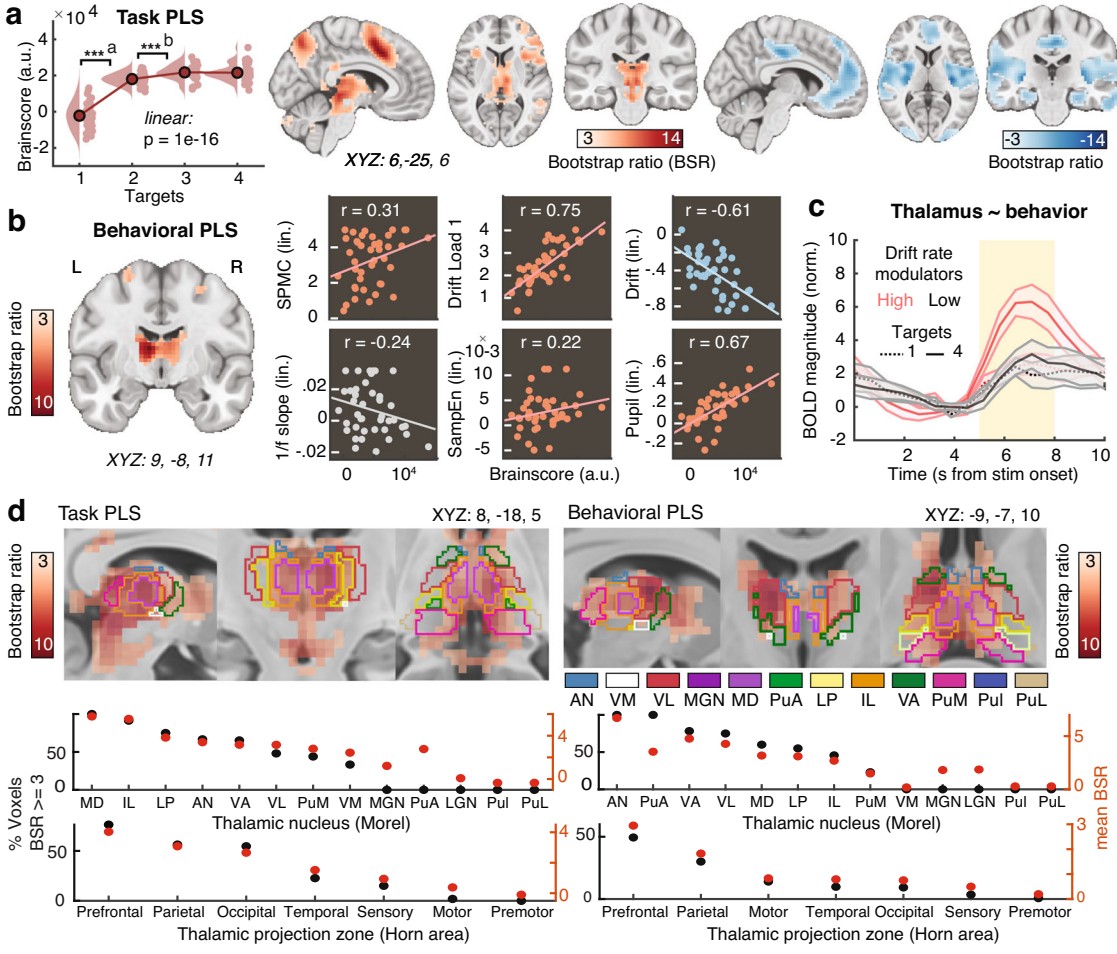

**Fig. 7 Upregulation of thalamic BOLD responses during stimulus processing is related to stronger excitability increases and better performance during upcoming decisions. a** Results from multivariate task partial least-squares (PLS) analysis investigating the relation of BOLD magnitude to attentional uncertainty. Data are within-subject centered for visualization (see "Methods"; $n = 42$ participants; ***a: $p = 1.9e-18$, ***b: $p = 8.7e-5$, linear: $b = 7491.1$, 95% CI = [6372.2, 8609.99], $t(41) = 13.52$). Panel **a** indicates $p$ values from two-tailed paired $t$-tests; pairwise comparisons were Benjamini-Hochberg-adjusted for multiple comparisons. Activity maps show positive (left) and negative (right) bootstrap ratios of LV1, thresholded at a bootstrap ratio of 3 ($p$ ~0.001). Supplementary Fig. 5a/b presents the full loading matrices for LV1 and LV2. **b** Results from behavioral PLS, probing the association between linear changes in BOLD magnitude with behavioral, electrophysiological, and pupillary changes under uncertainty ($n = 42$ participants). Supplementary Fig. 5c presents the LV's complete loadings. SPMC = spectral power modulation component. **c** Visualization of thalamic modulation with uncertainty, split between low- ($n = 21$) and high- ($n = 21$) behavioral drift modulators (mean ± SEM). The yellow shading indicates the approximate stimulus presentation period after accounting for the delay in the hemodynamic response function. Supplementary Fig. 5d plots all target conditions by group. **d** Thalamic expression pattern of the first task LV and the behavioral LV. Scatters below indicate the major nuclei and projection zones in which behavioral relations are maximally reliable. See "Methods" for abbreviations. Strongest expression is observed in antero-medial nuclei that project to fronto-parietal cortical targets. Source data are provided as a Source Data file.

decisions. We interpret these changes as a negative but necessary and adaptive consequence of the need to encode multiple relevant features for an eventual decision regarding a single target. Concurrently, BOLD activity increased in the dorsal frontoparietal network[33] (composed of the inferior frontal junction, inferior frontal gyrus, and posterior parietal cortex) and the midcingulo-insular network[33]. These cortical networks are associated with salience processing and are thought to establish the contextual relevance of environmental stimuli and to communicate this information to sensory cortex[19]. Accordingly, their BOLD activity often increases alongside multifaceted demands[34], further in line with increased mediofrontal theta engagement[29].

Besides such cortical responses at the group level, however, we noted that individual increases in cortical excitability, drift rates, and arousal were tracked primarily by the extent of thalamic signal elevation, dominantly in areas with fronto-parietal projections. While past work emphasized the thalamic relay of

peripheral information to cortex, recent theories highlight its dynamic involvement in cortical and cognitive function (for reviews see refs. [20,35,36]), with empirical support in humans[37–40], monkeys[41], and mice[23,42,43]. In particular, anterior and midline thalamic nuclei (such as the mediodorsal nucleus (MD), in which neuro-behavioral relations were maximal in our results) are implicated in establishing[43], sustaining[44], and switching[43,45–47] prefrontal rule representations depending on the active task context[43]. An intriguing possibility is that target uncertainty increases MD engagement to enable a more dynamic target selection among high-dimensional prefrontal feature representations[43,48,49]. Such proposal aligns with highest MD engagement during the integration of multiple cognitive demands[35,50], and MD lesions specifically impairing performance for larger stimulus sets[51] and more complex tasks[52]. As such, MD may play a crucial role in enabling flexible behaviors, particularly under uncertainty[35,50,53,54].

Such uncertainty modulation in anterior-medial thalamic regions complements the previously reported representation of perceptual precision or confidence in Pulvinar neurons[55,56]. Such "perceptual priors" may coordinate feature-selective information within and across visual cortex[20,41,55] once perceptual targets have been selected. Notably, the MAAT manipulates probe uncertainty, but not the sensory information (e.g., motion coherence) per se. It is thus an intriguing question whether a complementary manipulation of visual characteristics could dissociate MD and Pulvinar engagement in future work.

In sensory cortex, higher-order thalamic circuits can shape sensory excitability[23] via different thalamocortical activity modes[57]. In a "burst mode", thalamic nuclei elicit synchronous activity that can boost stimulus detection via non-linear gain in cortical responses, whereas spike activity during a "tonic mode" more faithfully tracks incoming signals[57]. Shifts from sparse bursts towards tonic activity may underlie attention-related increases in thalamic BOLD magnitude observed here and in previous task fMRI studies (e.g., ref. [38]), although further work needs to better elucidate the relation between thalamic transmission modes and BOLD responses (but see ref. [22]).

Associated with thalamic bursting, cortical alpha rhythms may control sensory gain via periodic fluctuations in excitability[5,7] that can signify rapid temporal imbalances between excitation and inhibition[58]. Supporting this notion, we observed a coupling between alpha phase and high-frequency power during stimulus processing, with participants engaging alpha rhythms most prevalently when prior cues afforded them a focus on single stimulus features (i.e., high available sensory evidence). Alpha rhythms have been consistently linked to the Pulvinar nucleus[41], which may leverage rhythms to establish and disband functional connectivity between visual and parietal cortex[55]. While the localization of effects within the thalamus remains challenging in BOLD signals[39], our results support a perspective in which alpha rhythms—shaped via thalamocortical circuits—dynamically extract relevant "bottom-up" sensory information[5] when contexts afford joint distractor suppression and target enhancement (such as the single-target condition in the present paradigm).

Complementing such selective gain control, overall increases in excitatory tone may serve multi-feature attention when only limited attentional guidance is available. Our results provide initial evidence that probe uncertainty transiently (a) desynchronizes alpha rhythms, (b) increases gamma power, and (c) elevates sample entropy while shallowing spectral slopes, a pattern that suggests increases in excitatory contributions to E/I mixture currents[13] and asynchronous neural firing[12]. Conceptually, elevated excitability during high probe uncertainty facilitates an efficient and rapid switching between parallel feature activations. Convergent with this idea, joint activation of neural populations coding multiple relevant features has been observed during multi-feature attention[11]. Furthermore, computational modeling indicates that E/I modulations in hierarchical networks optimally adjust multi-attribute choices[10]. Similar to our observation of enhanced excitability during probe uncertainty, Pettine et al.[10] found increases in excitatory tone optimal for a linear weighting of multiple features, whereas inhibitory engagement increased the gain for specific features during more difficult perceptual decisions. As discussed above, such inhibitory tuning may regulate selective target gains via alpha rhythms, in line with the presumed importance of inhibitory interneurons in alpha rhythmogenesis[6].

Finally, probe uncertainty increased phasic pupil diameter, with strong links to parallel adjustments in behavior, EEG-based excitability, and thalamic BOLD modulation. Fluctuations in pupil diameter provide a non-invasive proxy of noradrenergic drive[24]. As such, our results support neuromodulation as a potent regulator of excitability, at least in part via thalamic circuits[17,21]. Functionally, pupil diameter rises during states of heightened uncertainty (such as change points in dynamic environments[59,60]) and is accompanied by cortical desynchronization[31,59]. Our results extend those observations and suggest that neuromodulatory drive accompanies excitability increases, potentially to serve a more faithful processing of complex environments[17]. Indeed, multiple neuromodulators, prominently noradrenaline and acetylcholine, regulate thalamocortical excitability[2,3,17,61] and pupil responses[24], but may differentially serve perceptual sensitivity vs. specificity demands. Specifically, noradrenergic drive may increase sensitivity to external stimuli[17] by increasing E/I ratios[62–64], whereas cholinergic innervation might facilitate response selectivity[65] (but see ref. [61]). However, the functional separability of these modulators necessitates future work.

In summary, we report initial evidence that thalamocortical excitability adjustments guide human perception and decisions under uncertainty. Our results point to neuromodulatory changes regulated by the thalamus that trigger behaviorally relevant switches in cortical dynamics, from alpha-rhythmic gain control to increased tonic excitability once contexts require a more faithful processing of information-rich environments. Given that difficulties in dealing with uncertainty, neuro-sensory hyperexcitability, and deficient E/I control are all hallmarks of several clinical disorders (e.g., ref. [66]), we surmise that further research on individual differences in the modulation of contextual excitability may advance our understanding of cognitive flexibility in both healthy and disordered populations.

## Methods

**Sample**. Forty-seven healthy young adults (18–35 years, mean age = 25.8 years, SD = 4.6, 25 women) performed a dynamic visual attention task during 64-channel active scalp EEG acquisition, 42 of whom returned for a subsequent 3T fMRI session. Due to participant and scanner availability, the average span between EEG and MR testing sessions was 9.8 days (SD = 9.5 days). Participants were recruited from the participant database of the Max Planck Institute for Human Development, Berlin, Germany (MPIB). Participants were right-handed, as assessed with a modified version of the Edinburgh Handedness Inventory[67], and had normal or corrected-to-normal vision. Participants reported themselves to be in good health with no known history of neurological or psychiatric incidences, and were paid for their participation (10 € per hour). All participants gave written informed consent according to the institutional guidelines of the Deutsche Gesellschaft für Psychologie (DGPS) ethics board, which approved the study.

**Procedure: EEG session**. Participants were seated at a distance of 60 cm in front of a monitor in an acoustically and electrically shielded chamber with their heads placed on a chin rest. Following electrode placement, participants were instructed to rest with their eyes open and closed, each for 3 min. Afterwards, participants performed a standard Stroop task, followed by the visual attention task ("MAAT", see below) instruction and practice, the performance of the task and a second Stroop assessment (Stroop results are not reported here). Stimuli were presented on a 60 Hz 1920 × 1080p LCD screen (AG Neovo X24) using PsychToolbox 3.0.11[68–70]. The session lasted ~3 h. EEG was continuously recorded from 60 active (Ag/AgCl) electrodes using BrainAmp amplifiers (Brain Products GmbH, Gilching, Germany). Scalp electrodes were arranged within an elastic cap (EASYCAP GmbH, Herrsching, Germany) according to the 10% system[71], with the ground placed at AFz. To monitor eye movements, two additional electrodes were placed on the outer canthi (horizontal EOG) and one electrode below the left eye (vertical EOG). During recording, all electrodes were referenced to the right mastoid electrode, while the left mastoid electrode was recorded as an additional channel. Online, signals were digitized at a sampling rate of 1 kHz. In addition to EEG, we simultaneously tracked eye movements and assessed pupil diameter using EyeLink 1000+ hardware (SR Research, v.4.594) with a sampling rate of 1 kHz.

**Procedure: MRI session**. Forty-two participants returned for a second testing session that included structural and functional MRI assessments. First, participants took part in a short refresh of the visual attention task ("MAAT", see below) instructions and practiced the task outside the scanner. Then, participants were located in the TimTrio 3T scanner and were instructed in the button mapping. We collected the following sequences: T1w, task (4 runs), T2w, resting state, DTI, with a 15 min out-of-scanner break following the task acquisition. The session lasted ~3 h. Whole-brain task fMRI data (4 runs á ~11.5 min, 1066 volumes per run) were collected via a 3T Siemens TrioTim MRI system (Erlangen, Germany) using a

multi-band EPI sequence (factor 4; TR = 645 ms; TE = 30 ms; flip angle 60°; FoV = 222 mm; voxel size 3 × 3 × 3 mm; 40 transverse slices. The first 12 volumes (12 × 645 ms = 7.7 s) were removed to ensure a steady state of tissue magnetization (total remaining volumes = 1054 per run). A T1-weighted structural scan was also acquired (MPRAGE: TR = 2500 ms; TE = 4.77 ms; flip angle 7°; FoV = 256 mm; voxel size 1 × 1 × 1 mm; 192 sagittal slices). A T2-weighted structural scan was also acquired (GRAPPA: TR = 3200 ms; TE = 347 ms; FoV = 256 mm; voxel size 1 × 1 × 1 mm; 176 sagittal slices).

**The Multi-Attribute Attention Task**. We designed a task to parametrically control top-down attention to multiple feature dimensions, in the absence of systematic variation in bottom-up visual stimulation (see Fig. 1). Participants were shown a dynamic square display that jointly consisted of four attributes: color (red/ green), movement direction (left, right), size (small, large), and saturation (low, high). The task incorporates features from random dot motion tasks which have been extensively studied in both animal models[19,25,72] and humans[73,74]. Following the presentation of these displays, a probe queried the prevalence of one of the four attributes in the display (e.g., whether the display comprised a greater proportion of either smaller or larger squares). Prior to stimulus onset, valid cue presentation informed participants about the active feature set, out of which one feature would be chosen as the probe. We parametrically manipulated uncertainty regarding the upcoming probe by systematically varying both the number and type of relevant features in the display.

The difficulty of each feature was determined by (a) the fundamental feature difference between the two alternatives and (b) the sensory evidence for each alternative in the display. For (a) the following values were used: high (RGB: 128, 255, 0) and low saturation green (RGB: 192, 255, 128) and high (RGB: 255, 0, 43) and low saturated red (RGB: 255, 128, 149) for color and saturation, 5 and 8 pixels for size differences and a coherence of 0.2 for directions. For (b) the proportion of winning to losing option (i.e., sensory evidence) was chosen as follows: color: 60/ 40; direction: 80/20; size: 65/35; luminance: 60/40. Parameter difficulty was established in a pilot population, with the aim to produce above-chance accuracy for individual features.

The experiment consisted of four runs of ~10 min, each consisting of eight blocks of eight trials (i.e., a total of 32 trial blocks; 256 trials). The size and constellation of the cue set was held constant within eight-trial blocks to reduce set switching and working memory demands. Each trial was structured as follows: cue onset during which the relevant targets were centrally presented (1 s), fixation phase (2 s), dynamic stimulus phase (3 s), probe phase (incl. response; 2 s); ITI (un-jittered; 1.5 s). At the onset of each block, the valid cue (attentional target) set was presented for 5 s. At the offset of each block, participants received feedback for 3 s. The four features with two options each spanned a constellation of 16 distinct stimulus combinations, of which presentation frequency was matched within participants. The size and type of cue set was pseudo-randomized, such that every size and constellation of the cue set was presented across blocks. Within each run of four blocks, every set size was presented once, but never directly following a block of the same set size. In every block, each feature in the active set acted as a probe in at least one trial. Moreover, any attribute equally often served as a probe across all blocks. Winning options for each feature were balanced across trials such that correct responses were equally distributed to the left and rigth button across the experiment. To retain high motivation during the task and encourage fast and accurate responses, we instructed participants that one response would randomly be drawn at the end of each block; if this response was correct and faster than the mean RT during the preceding block, they would earn a reward of 20 cents. However, we pseudo-randomized feedback such that all participants received a fixed payout of 10 € per session. This extra money was paid in addition to the participation fee at the end of the second session, at which point participants were debriefed.

**Behavioral estimates of probe-related decision processes.** Sequential sampling models, such as the drift-diffusion model (DDM[26]), have been used to characterize evolving perceptual decisions in 2-alternative forced choice (2AFC) random dot motion tasks[73], where the evolving decision relates to overt stimulus dynamics. In contrast to such applications, evidence integration here is tied to eidetic memory traces following the probe onset, similar to applications during memory retrieval[75] or probabilistic decision making[76]. Here, we estimated individual evidence integration parameters within the HDDM 0.6.0 toolbox[77] to profit from the large number of participants that can establish group priors for the relatively sparse within-subject data. Independent models were fit to data from the EEG and the fMRI session to allow reliability assessments of individual estimates. Premature responses faster than 250 ms were excluded prior to modeling, and the probability of outliers was set to 5%. In total, 7000 Markov-Chain Monte Carlo samples were sampled to estimate parameters, with the first 5000 samples being discarded as burn-in to achieve convergence. We judged convergence for each model by visually assessing both Markov chain convergence and posterior predictive fits. Individual estimates were averaged across the remaining 2000 samples for follow-up analyses.

We fitted data to correct and incorrect RTs (termed "accuracy coding" in Wiecki et al.[77]). To explain differences in decision components, we compared four separate models. In the "full model", we allowed the following parameters to vary between conditions: (i) the mean drift rate across trials, (ii) the threshold

separation between the two decision bounds, (iii) the NDT, which represents the summed duration of sensory encoding and response execution. In the remaining models, we reduced model complexity, by only varying (a) drift, (b) drift + threshold, or (c) drift + NDT, with a null model fixing all three parameters. For model comparison, we first used the Deviance Information Criterion (DIC) to select the model which provided the best fit to our data. The DIC compares models on the basis of the maximal log-likelihood value, while penalizing model complexity. The full model provided the best fit to the empirical data based on the DIC index (Supplementary Fig. 1c) in both the EEG and the fMRI session. However, although this model did indicate an increase in decision thresholds (i.e., boundary separation), there was no equivalent effect noted in the electrophysiological data (Supplementary Fig. 1d). We therefore fixed the threshold parameter across conditions, in line with previous work constraining DDM model parameters on the basis of electrophysiological evidence[28].

**EEG preprocessing**. Preprocessing and analysis of EEG data were conducted with the FieldTrip toolbox (v.20170904)[78] and using custom-written MATLAB (The MathWorks Inc., Natick, MA, USA) code. Offline, EEG data were filtered using a fourth-order Butterworth filter with a pass-band of 0.5–100 Hz. Subsequently, data were downsampled to 500 Hz and all channels were re-referenced to mathematically averaged mastoids. Blink, movement and heart-beat artifacts were identified using independent component analysis (ICA[79]) and removed from the signal. Artifact-contaminated channels (determined across epochs) were automatically detected using (a) the FASTER algorithm[80] and by (b) detecting outliers exceeding three standard deviations of the kurtosis of the distribution of power values in each epoch within low (0.2–2 Hz) or high (30–100 Hz) frequency bands, respectively. Rejected channels were interpolated using spherical splines[81]. Subsequently, noisy epochs were likewise excluded based on a custom implementation of FASTER and on recursive outlier detection. Finally, recordings were segmented to participant cues to open their eyes and were epoched into non-overlapping 3 s pseudo-trials. To enhance spatial specificity, scalp current density estimates were derived via fourth-order spherical splines[81] using a standard 1005 channel layout (conductivity: 0.33 s/m; regularization: $1^{-05}$; 14th degree polynomials).

**Electrophysiological estimates of probe-related decision processes**
*Centro-parietal positivity*. The CPP is an electrophysiological signature of internal evidence-to-bound accumulation[28,73,82]. We probed the task modulation of this established signature and assessed its convergence with behavioral parameter estimates. To derive the CPP, preprocessed EEG data were low-pass filtered at 8 Hz with a sixth-order Butterworth filter to exclude low-frequency oscillations, epoched relative to response, and averaged across trials within each condition. In accordance with the literature, this revealed a dipolar scalp potential that exhibited a positive peak over parietal channel POz (see Fig. 2). We temporally normalized individual CPP estimates to a condition-specific baseline during the final 250 ms preceding probe onset. As a proxy of evidence drift rate, CPP slopes were estimates via linear regression from −250 to −100 ms surrounding response execution, while the average CPP amplitude from −50 to 50 ms served as an indicator of decision thresholds (i.e., boundary separation; e.g., ref. [28]).

To investigate whether a similar "ramping" potential was observed during stimulus presentation, we aligned data to stimulus onset and temporally normalized signals to the condition-specific signal during the final 250 ms prior to stimulus onset. During stimulus presentation, no "ramp"-like signal or load modulation was observed at the peak CPP channel. This suggests that immediate choice requirements were necessary for the emergence of the CPP, although prior work has shown the CPP to be independent of explicit motor requirements[82].

Finally, we assessed whether differences between probed stimulus attributes could account for load-related CPP changes (Supplementary Fig. 2e–g). For this analysis, we selected trials separately by condition and probed attribute. Note that for different probes (but not cues), trials were uniquely associated with each feature and trial counts were approximately matched across conditions. We explored differences between different conditions via paired t-tests. To assess load effects on CPP slopes and thresholds as a function of probed attribute, we calculated first-level load effects by means of a linear model and assessed their difference from zero via paired t-tests.

*Contralateral mu–beta*. Decreases in contralateral mu-beta power provide a complementary, effector-specific signature of evidence integration[28,83]. We estimated mu-beta power using seven-cycle wavelets for the 8–25 Hz range with a step size of 50 ms. Spectral power was time-locked to probe presentation and response execution. We re-mapped channels to describe data recorded contra- and ipsilateral to the executed motor response in each trial, and averaged data from those channels to derive grand average mu-beta time courses. Individual average mu-beta time series were baseline-corrected using the −400 to −200 ms time window prior to probe onset, separately for each condition. For contralateral motor responses, remapped sites C3/5 and CP3/CP5 were selected based on the grand average topography for lateralized response executions (see inset in Supplementary Fig. 2a). As a proxy of evidence drift rate, mu-beta slopes were estimates via linear regression from −250 to −50 ms prior to response execution, while the average power −50 to 50 ms served as an indicator of decision thresholds (e.g., ref. [28]).

### Electrophysiological indices of top-down modulation during sensation

*Low-frequency alpha and theta power.* We estimated low-frequency power via a seven-cycle wavelet transform, applied at linearly spaced center frequencies in 1 Hz steps from 2 to 15 Hz. The step size of estimates was 50 ms, ranging from −1.5 s prior to cue onset to 3.5 s following stimulus offset. Estimates were log10-transformed at the single trial level[84], with no explicit baseline.

*High-frequency gamma power.* Gamma responses were estimated using multi-tapers (five tapers; discrete prolate spheroidal sequences) with a step size of 200 ms, a window length of 400 ms, and a frequency resolution of 2.5 Hz. The frequency range covered frequencies between 45 and 90 Hz, with spectral smoothing of 8 Hz. Estimates were log10-transformed at the single trial level. We normalized individual gamma-band responses via single-trial $z$-normalization to decrease sensitivity to non-neural sources[85]. In particular, for each frequency, we subtracted single-trial power −700 to −100 ms prior to stimulus onset and divided by the standard deviation of power values across trials during the same period. This produces normalized values that weight stimulus-induced gamma power relative to (presumed non-neural) gamma power variation in the absence of visual stimulation. Finally, to account for baseline shifts during the pre-stimulus period, we subtracted condition-wise averages during the same baseline period.

*Multivariate assessment of spectral power changes with stimulus onset and uncertainty.* To determine changes in spectral power upon stimulus onset, and during stimulus presentation with load, we entered individual power values into multivariate partial least-squares (PLS) analyses (see section "Multivariate partial least-squares analyses") using the MEG-PLS toolbox v2.02b[86]. We concatenated low-(2–15 Hz) and high-frequency (45–90 Hz) power matrices to assess joint changes in the PLS models. To examine a multivariate contrast of spectral changes upon stimulus onset (averaged across conditions) with spectral power in the pre-stimulus baseline period, we performed a task PLS on data ranging from 500 ms pre-stim to 500 ms post-stim. Temporal averages from −700 to −100 ms pre-stimulus onset were subtracted as a baseline. To assess power changes as a function of probe uncertainty, we segmented the data from 500 ms post stim onset to stimulus offset (to exclude transient evoked onset responses) and calculated a task PLS concerning the relation between experimental uncertainty conditions and time–space–frequency power values. As a control, we performed a behavioral PLS analysis to assess the relevance of individual frequency contributions to the behavioral relation. For this analysis, we computed linear slopes relating power to target load for each time–frequency point at the first (within-subject) level, which were subsequently entered into the second-level PLS analysis. On the behavioral side, we assessed both linear changes in pupil diameter, as well as drift rates in the single-target condition and linear decreases in drift rate under uncertainty. Finally, spontaneous fluctuations in pre-stimulus power have been linked to fluctuations in cortical excitability[87,88]. We thus probed the role of upcoming processing requirements on pre-stimulus oscillations, as well as the potential relation to behavioral outcomes using task and behavioral PLS analyses. The analysis was performed as described above but was restricted to time points occurring during the final second prior to stimulus onset.

*Steady-state visual-evoked potential.* The SSVEP characterizes the phase-locked, entrained visual activity (here 30 Hz) during dynamic stimulus updates (e.g., ref. [89]). These features differentiate it from induced broadband activity or muscle artifacts in similar frequency bands. We used these properties to normalize individual single-trial SSVEP responses prior to averaging: (a) we calculated an FFT for overlapping 1 s segments with a step size of 100 ms (Hanning-based multitaper) and averaged them within each load condition; (b) we spectrally normalized 30 Hz estimates by subtracting the average of estimates at 28 and 32 Hz, effectively removing broadband effects (i.e., aperiodic slopes), and; (c) we subtracted a temporal baseline −700 to −100 ms prior to stimulus onset. Linear load effects on SSVEPs were assessed by univariate cluster-based permutation tests on channel × time data (see "Univariate cluster-based permutation analyses").

*Time-resolved sample entropy.* Sample entropy[32] quantifies the irregularity of a time series of length $N$ by assessing the conditional probability that two sequences of $m$ consecutive data points will remain similar when another sample $(m + 1)$ is included in the sequence (for a visual example see Fig. 1A in ref. [15]). Sample entropy is defined as the inverse natural logarithm of this conditional similarity: $\mathrm{SampEn}(m, r, N) = -\log\left(\frac{p^{m+1}(r)}{p^m(r)}\right)$. The similarity criterion $(r)$ defines the tolerance within which two points are considered similar and is defined relative to the standard deviation (~variance) of the signal (here set to $r = 0.5$). We set the sequence length $m$ to 2, in line with previous applications[15]. An adapted version of sample entropy calculations implemented in the mMSE toolbox (available from https://github.com/LNDG/mMSE) was used[15,90,91], wherein entropy is estimated across discontinuous data segments to provide time-resolved estimates. The estimation of scale-wise entropy across trials allows for an estimation of coarse scale entropy also for short time-bins (i.e., without requiring long, continuous signals), while quickly converging with entropy estimates from continuous recordings[90]. To remove the influence of posterior-occipital low-frequency rhythms on entropy estimates, we notch-filtered the 8–15 Hz alpha band using sixth-order Butterworth

filter prior to the entropy calculation[15]. Time-resolved entropy estimates were calculated for 500 ms windows from −1 s pre-stimulus to 1.25 s post-probe with a step size of 150 ms. As entropy values are implicitly normalized by the variance in each time bin via the similarity criterion, no temporal baselining was required. Linear load effects on entropy were assessed by univariate cluster-based permutation tests on channel × time data (see "Univariate cluster-based permutation analyses").

*Aperiodic (1/f) slopes.* The aperiodic $1/f$ slope of neural recordings is closely related to the sample entropy of broadband signals[15] and has been suggested as a proxy for "cortical excitability" and excitation–inhibition balance[13]. Spectral estimates were computed by means of a fast Fourier transform (FFT) over the final 2.5 s of the presentation period (to exclude onset transients) for 41 logarithmically spaced frequencies between 2 and 64 Hz (Hanning-tapered segments zero-padded to 10 s) and subsequently averaged. Spectral power was log10-transformed to render power values more normally distributed across participants. Power spectral density (PSD) slopes were derived by linearly regressing log10-transformed power values on log10-transformed frequencies. The spectral range from 7 to 13 Hz was excluded from the background fit to exclude a bias by the narrowband alpha peak[15] and thus to increase the specificity to aperiodic variance. Linear load effects on $1/f$ slopes were assessed by univariate cluster-based permutation tests on channel data (see "Univariate cluster-based permutation analyses").

*Rhythm-specific estimates.* Spectral power estimates conflate rhythmic events with aperiodicity in time, space, and magnitude[92]. Given that we observed changes in aperiodic slopes, we verified that observed narrowband effects in the theta and alpha band describe narrowband changes in rhythmicity. For this purpose, we identified single-trial spectral events using the extended BOSC (eBOSC) method[92–94]. In short, this method identifies stereotypic "rhythmic" events at the single-trial level, with the assumption that such events have significantly higher power than the 1/f background and occur for a minimum number of cycles at a particular frequency. This procedure dissociates narrowband spectral peaks from the aperiodic background spectrum. Here, we used a three-cycle threshold during detection, while defining the power threshold as the 95th percentile above the individual background power. A five-cycle wavelet was used to provide the time–frequency transformations for 49 logarithmically spaced center frequencies between 1 and 64 Hz. Rhythmic episodes were detected as described in ref. [92]. Prior to fitting 1/f slopes, the most dominant individual rhythmic alpha peak between 8 and 15 Hz was removed, as well as the 28–32 Hz range, to exclude the SSVEP. Detection of episodes was restricted to the time of stimulus presentation, excluding the first 500 ms to reduce residual pre-stimulus activity and onset transients. Within each participant and channel, the duration and SNR of individual episodes with a mean frequency between 4 and 8 Hz (theta) and 8 and 15 Hz (alpha) were averaged across trials. Effects of target number were assessed within the averaged spatial clusters indicated in Fig. 3 by means of paired $t$-tests.

**Alpha–gamma PAC.** We assessed alpha-phase-to-gamma-amplitude coupling to assess the extent of phasic modulation of gamma power within the alpha band. As phase information is only interpretable during the presence of a narrowband rhythm[95], we focused our main analysis on 250 ms time segments following the estimated onset of a rhythm in the 8–15 Hz alpha range (see section "Rhythm-specific estimates"; Fig. 4a). This time window ensured that segments fulfilled the three-cycle criterion imposed during eBOSC rhythm detection to ensure that a rhythm was present. We selected three occipital channels with maximal gamma power (O1, O2, Oz; shown in Fig. 4a) and pooled detected alpha episodes across these channels. We pooled data across load conditions, as we observed no consistent PAC within individual load conditions, perhaps due to low episode counts. To derive the alpha carrier phase, we band-pass filtered signals in the 8–15 Hz band, and estimated the analytic phase time series via Hilbert transform. For the amplitude of modulated frequencies, we equally applied band-pass filters from 40 to 150 Hz (step size: 2 Hz), with adaptive bandwidths (±20% of center frequency). Filtering was implemented using MATLAB's acausal filtfilt() routine using linear finite impulse response filters, with an adaptive filter order set as three times the ratio of the sampling frequency to the low-frequency cutoff[96]. For each applied bandpass filter, we removed 250 ms at each edge to avoid filter artifacts. For each frequency, narrowband signals were $z$-scored to normalize amplitudes across frequencies, and absolute values of the Hilbert-derived complex signal were squared to produce instantaneous power time series. We estimated the MI between the 8 and 15 Hz phase and high-frequency power via normalized entropy[96] using 16 phase bins. Power estimates were normalized by dividing the bin-specific power by the sum of power across bins. To make MI estimates robust against random coupling, we estimated MI for 1000 surrogates, which shuffled the trial association of phase and amplitude information. We subtracted the mean surrogate MI value from the original MI index for a final, surrogate-normalized MI estimate. The resulting MI estimates across frequencies were then subjected to a cluster-based permutation test to assess significant clusters from zero using paired $t$-tests. For Fig. 4b, we followed the procedure by Canolty et al.[97]. Alpha troughs were identified as local minima of phases $<[-\mathrm{pi} + 0.01]$. For visualization, data were averaged across center frequencies from 80 to 150 Hz, as significant coupling

overlapped with this range. We performed identical analyses for the 250 ms periods prior to rhythm onset (gray shading in Fig. 4a) as a control condition.

**Analyses of pupil diameter.** Pupil diameter was recorded during the EEG session using EyeLink 1000 at a sampling rate of 1000 Hz and was analyzed using FieldTrip and custom-written MATLAB scripts. Blinks were automatically indicated by the EyeLink software (version 4.40). To increase the sensitivity to periods of partially occluded pupils or eye movements, the first derivative of eye-tracker-based vertical eye movements was calculated, z-standardized, and outliers ≥3 STD were removed. We additionally removed data within 150 ms preceding or following indicated outliers. Finally, missing data were linearly interpolated, and data were epoched to 3.5 s prior to stimulus onset to 1 s following stimulus offset. We quantified phasic arousal responses via the first temporal derivative (i.e. rate of change) of pupil diameter traces as this measure (i) has higher temporal precision and (ii) has been more strongly associated with noradrenergic responses than the overall response[24]. We downsampled pupil time series to 200 Hz. For visualization, but not statistics, we smoothed pupil traces using a moving average median of 200 ms. We statistically assessed a linear load effect using a cluster-based permutation test on the 1D pupil traces (see "Univariate cluster-based permutation analyses"). For post hoc assessments, we extracted the median pupil derivative during the first 1.5 s following stimulus onset.

### fMRI-based analyses

*Preprocessing of functional MRI data.* fMRI data were preprocessed with FSL 5 (RRID:SCR_002823)[98,99]. Pre-processing included motion correction using McFLIRT, smoothing (7 mm), and high-pass filtering (0.01 Hz) using an eighth-order zero-phase Butterworth filter applied using MATLAB's filtfilt function. We registered individual functional runs to the individual, ANTs brain-extracted T2w images (6 DOF), to T1w images (6 DOF) and finally to 3 mm standard space (ICBM 2009c MNI152 nonlinear symmetric)[100] using nonlinear transformations in ANTs 2.1.0 (ref. [101]) (for one participant, no T2w image was acquired and 6 DOF transformation of BOLD data was preformed directly to the T1w structural scan). We then masked the functional data with the ICBM 2009c GM tissue prior (thresholded at a probability of 0.25), and detrended the functional images (up to a cubic trend) using SPM12's spm_detrend.

We also used a series of extended preprocessing steps to further reduce potential non-neural artifacts[102,103]. Specifically, we examined data within-subject, within-run via spatial independent component analysis (ICA) as implemented in FSL-MELODIC[104]. Due to the high multiband data dimensionality in the absence of low-pass filtering, we constrained the solution to 30 components per participant. Noise components were identified according to several key criteria: (a) spiking (components dominated by abrupt time series spikes); (b) motion (prominent edge or "ringing" effects, sometimes [but not always] accompanied by large time series spikes); (c) susceptibility and flow artifacts (prominent air-tissue boundary or sinus activation; typically represents cardio/respiratory effects); (d) white matter (WM) and ventricle activation[105]; (e) low-frequency signal drift[106]; (f) high power in high-frequency ranges unlikely to represent neural activity (≥75% of total spectral power present above 0.10 Hz); and (g) Spatial distribution ("spotty" or "speckled" spatial pattern that appears scattered randomly across ≥25% of the brain, with few if any clusters with ≥80 contiguous voxels). Examples of these various components we typically deem to be noise can be found in ref. [107]. By default, we utilized a conservative set of rejection criteria; if manual classification decisions were challenging due to mixing of "signal" and "noise" in a single component, we generally elected to keep such components. Three independent raters of noise components were utilized; >90% inter-rater reliability was required on separate data before denoising decisions were made on the current data. Components identified as artifacts were then regressed from corresponding fMRI runs using the regfilt command in FSL.

To reduce the influence of motion and physiological fluctuations, we regressed FSL's 6 DOF motion parameters from the data, in addition to average signal within white matter and CSF masks. Masks were created using 95% tissue probability thresholds to create conservative masks. Data and regressors were demeaned and linearly detrended prior to multiple linear regression for each run. To further reduce the impact of potential motion outliers, we censored significant DVARS outliers during the regression as described by Power et al.[108]. In particular, we calculated the "practical significance" of DVARS estimates and applied a threshold of 5 (ref. [109]). The regression-based residuals were subsequently spectrally interpolated during DVARS outliers as described in refs. [108,110]. BOLD analyses were restricted to participants with both EEG and MRI data available (N = 42).

*First-level analysis: univariate beta weights for load conditions.* We conducted a first-level analysis using SPM12 to identify beta weights for each load condition separately. Design variables included stimulus presentation by load (4 volumes; parametrically modulated by sequence position), onset cue (no mod.), and probe (2 volumes, parametric modulation by RT). Design variables were convolved with a canonical HRF, including its temporal derivative as a nuisance term. Nuisance regressors included 24 motion parameters[111] as well as continuous DVARS estimates. Autoregressive modeling was implemented via FAST. For each load condition, output beta images were averaged across runs.

*Second-level analysis: multivariate modulation of BOLD responses.* We investigated the multivariate modulation of the BOLD response at the second level using PLS analyses (see section "Multivariate partial least-squares analyses"). Specifically, we probed the relationship between voxel-wise first-level beta weights and probe uncertainty (i.e., task load level) within a task PLS. Next, we assessed the relationship between task-related BOLD signal changes and interindividual differences in the joint modulation of decision processes, cortical excitability, and pupil modulation by means of a behavioral PLS. For this, we first calculated linear slope coefficients for voxel-wise beta estimates. Then, we included behavioral variables, including HDDM parameter estimates in the single-target condition, as well as linear changes with load, individual linear condition modulation of the following variables: multivariate spectral power, pupil dilation, 1/f modulation, and entropy residuals. Prior to these covariates in the model, we visually assessed whether the distribution of linear changes variables was approximately Gaussian. In the case of outliers (as observed for the SPMC, 1/f slopes, and entropy), we winsorized values to the 95th percentile. For visualization, spatial clusters were defined based on a minimum distance of 10 mm, and by exceeding a size of 25 voxels. We identified regions associated with peak activity based on cytoarchitectonic probabilistic maps implemented in the SPM Anatomy Toolbox (Version 2.2c)[112]. If no assignment was found, the most proximal assignment within the cluster to the coordinates reported in Supplementary Table 1 was reported.

*Temporal dynamics of thalamic engagement.* To visualize the modulation of thalamic activity by load, we extracted signals within a binary thalamic mask extracted from the Morel atlas[113], including all subdivisions. Preprocessed BOLD timeseries were segmented into trials, spanning the period from the stimulus onset to the onset of the feedback phase. Given a time-to-peak of a canonical hemodynamic response function (HRF) between 5 and 6 s, we designated the 3 s interval from 5 to 8 s following the stimulus onset trigger as the stimulus presentation interval, and the 2 s interval from 3 to 5 s as the fixation interval, respectively. Single-trial time series were then temporally normalized to the temporal average during the approximate fixation interval. To visualize inter-individual differences in thalamic engagement (see Fig. 7c), we performed a median split across participants based on their individual drift modulation.

*Thalamic loci of behavioral PLS.* To assess the thalamic loci of most reliable behavioral relations (Fig. 7d), we assessed bootstrap ratios within two thalamic masks. First, for nucleic subdivisions, we used the Morel parcellation scheme as consolidated and kindly provided by Hwang et al.[39] for 3 mm data at 3 T field strength. The abbreviations are as follows: AN: anterior nucleus; VM: ventromedial; VL: ventrolateral; MGN: medial geniculate nucleus; LGN: lateral geniculate nucleus; MD: mediodorsal; PuA: anterior pulvinar; LP: lateral-posterior; IL: intra-laminar; VA: ventral-anterior; PuM: medial pulvinar; Pul: pulvinar proper; PuL: lateral pulvinar. Second, to assess cortical white-matter projections we considered the overlap with seven structurally-derived cortical projection zones suggested by Horn and Blankenburg[114], which were derived from a large adult sample (N = 169). We binarized continuous probability maps at a relative 75% threshold of the respective maximum probability, and re-sliced masks to 3 mm (ICBM 2009c MNI152).

### Statistical analyses

*Assessment of covarying load effect magnitudes between measures.* To assess a linear modulation of dependent variables, we calculated first-level beta estimates for the effect of load ($y = \text{intercept} + \beta * \text{LOAD} + e$) and assessed the slope difference from zero at the group level using two-sided paired $t$-tests. We performed post hoc comparisons between adjacent load conditions using two-tailed paired $t$-tests, and adjusted $p$ values according to the Benjamini–Hochberg false discovery rate procedure[115] to account for multiple comparisons. We assessed the relation of individual load effects between measures of interest by means of partial repeated measures correlations, which we implemented in R 4.0.3[116]. In a simplified form, repeated measured correlation[117] fits a linear model between two variables $x1$ and $x2$ of interest, while controlling for repeated assessments within participants

$$[x1 \sim 1 + \beta1^* \text{ID} + \beta2^* x2 + e]. \tag{1}$$

Crucially, to exclude bivariate relations that exclusively arise from the overall main effect of number of targets, we added target load as an additional categorical covariate

$$[x1 \sim 1 + \beta1^* \text{ID} + \beta2^* \text{LOAD} + \beta3^* x2 + e] \tag{2}$$

to remove group condition means. Resulting estimates characterize the group-wise coupling in the (zero-centered) magnitude of changes between the DV and the IV across the four load levels. To identify the directionality of the coupling, we assessed the direction of main effects for $x1$ and $x2$. We statistically compared this model to a null model without the term of interest

$$[x1 \sim 1 + \beta1^* \text{ID} + \beta2^* \text{LOAD} + e] \tag{3}$$

to assess statistical significance. We report the bivariate residual effect size by assessing the square root of partial eta squared. We extend this model with additional beta*covariate terms when reporting control for additional covariates.

*Within-subject centering*. To visually emphasize effects within participants, we use within-subject centering across repeated measures conditions by subtracting individual cross-condition means and adding global group means. For these visualizations, only the mean of the dependent values directly reflects the original units of measurement, as individual data points by construction do not reflect between-subject variation averaged across conditions. This procedure equals the creation of within-subject standard errors[118]. Within-subject centering is exclusively used for display and explicitly noted in the respective legends, but is not required for statistical calculations (i.e., our model resolves these intended estimates directly).

*Univariate cluster-based permutation analyses*. For data with a low-dimensional structure (e.g., based on a priori averaging or spatial cluster assumptions), we used univariate cluster-based permutation analyses (CBPAs) to assess significant modulations by target load or with stimulus onset. These univariate tests were performed by means of dependent samples *t*-tests, and cluster-based permutation tests[119] were performed to control for multiple comparisons. Initially, a clustering algorithm formed clusters based on significant *t*-tests of individual data points ($p <$ 0.05, two sided; cluster entry threshold) with the spatial constraint of a cluster covering a minimum of three neighboring channels. Then, the significance of the observed cluster-level statistic (based on the summed *t* values within the cluster) was assessed by comparison to the distribution of all permutation-based cluster-level statistics. The final cluster *p* value that we report in all figures was assessed as the proportion of 1000 Monte Carlo iterations in which the cluster-level statistic was exceeded. Cluster significance was indicated by *p* values below 0.025 (two-sided cluster significance threshold).

*Multivariate partial least-squares analyses*. For data with a high-dimensional structure, we performed multivariate partial-least squares analyses[30,120]. To assess main effect of probe uncertainty or stimulus onset, we performed Task PLS analyses. Task PLS begins by calculating a between-subject covariance matrix (COV) between conditions and each neural value (e.g., time-space-frequency power), which is then decomposed using singular value decomposition (SVD). This yields a left singular vector of experimental condition weights (*U*), a right singular vector of brain weights (*V*), and a diagonal matrix of singular values (*S*). Task PLS produces orthogonal latent variables (LVs) that reflect optimal relations between experimental conditions and the neural data. To examine multivariate relations between neural data and other variables of interest, we performed behavioral PLS analyses. This analysis initially calculates a between-subject correlation matrix (CORR) between (1) each brain index of interest (e.g., spectral power, first-level BOLD beta values) and (2) a second "behavioral" variable of interest (note that although called behavioral, this variable can reflect any variable of interest, e.g., behavior, pupil dilation, spectral power). CORR is then decomposed using singular value decomposition (SVD): $\text{SVD}_{\text{CORR}} = USV'$, which produces a matrix of left singular vectors of cognition weights (*U*), a matrix of right singular vectors of brain weights (*V*), and a diagonal matrix of singular values (*S*). For each LV (ordered strongest to weakest in *S*), a data pattern results which depicts the strongest available relation between brain data and other variables of interest. Significance of detected relations of both PLS model types was assessed using 1000 permutation tests of the singular value corresponding to the LV. A subsequent bootstrapping procedure indicated the robustness of within-LV neural saliences across 1000 resamples of the data[121]. By dividing each brain weight (from *V*) by its bootstrapped standard error, we obtained "bootstrap ratios" (BSRs) as normalized robustness estimates. We generally thresholded BSRs at values of ±3.00 (~99.9% confidence interval). We also obtained a summary measure of each participant's robust expression of a particular LV's pattern (a within-person "brain score") by multiplying the vector of brain weights (*V*) from each LV by each participant's vector of neural values (*P*), producing a single within-subject value: Brain score $= VP'$.

**Reporting summary**. Further information on research design is available in the Nature Research Reporting Summary linked to this article.

## Data availability
Primary EEG, fMRI, and behavioral data are available from https://osf.io/ug4b8/ (https://doi.org/10.17605/OSF.IO/UG4B8). Structural MRI data are exempt from public sharing according to informed consent. All data are available from the corresponding authors upon reasonable request. Source data are provided with this paper.

## Code availability
Experiment code is available from https://git.mpib-berlin.mpg.de/LNDG/multi-attribute-task. Analysis code is available from https://git.mpib-berlin.mpg.de/LNDG/stateswitch.

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

## Acknowledgements

This study was conducted within the 'Lifespan Neural Dynamics Group' at the Max Planck UCL Centre for Computational Psychiatry and Ageing Research in the Max Planck Institute for Human Development (MPIB) in Berlin, Germany. DDG was supported by an Emmy Noether Programme grant from the German Research Foundation. D.D.G. and U.L. were partially supported by the Max Planck UCL Centre for Computational Psychiatry and Ageing Research, and U.L. acknowledges financial support from the Intramural Innovation Fund of the Max Planck Society. J.Q.K. is a pre-doctoral fellow supported by the International Max Planck Research School on Computational Methods in Psychiatry and Ageing Research (IMPRS COMP2PSYCH). The participating institutions are the Max Planck Institute for Human Development, Berlin, Germany, and University College London, London, UK. For more information, see https://www.mps-ucl-centre.mpg.de/en/comp2psych. The funders had no role in study design, data collection and analysis, decision to publish, or preparation of the manuscript. We thank our research assistants and participants for their contributions to the present work, Alistair Perry for assistance in fMRI preprocessing, and Steffen Wiegert for organizational support.

## Author contributions

J.Q.K.: conceptualization, methodology, investigation, software, formal analysis, visualization, writing—original draft, writing—review and editing, validation, data curation; U.L.: conceptualization, resources, writing—review and editing, supervision, funding acquisition; D.D.G.: conceptualization, methodology, software, resources, writing—review and editing, supervision, project administration, funding acquisition.

## Funding

## Competing interests

The authors declare no competing interests.
