## [Peer Review File · Nature Communications]

REVIEWER COMMENTS

Reviewer #1 (Remarks to the Author):

The manuscript by Kosciessa et al utilizes a combination of EEG, fMRI and a perceptual decision making task to identify shifts in cortical activity patterns that underlie perceptual uncertainty on one end and the involvement of the thalamus in potentially mediating these shifts, on another. The study is quite topical and of broad relevance. I really enjoyed reading the manuscript and very much appreciate the rigor and thoroughness here.

Below are a few points of clarification that should help make the manuscript more accessible and easier to read:

1. The premise of the author's hypothesis that heightened uncertainty would shift cortical state is never spelled out. I recognize that, after going through the manuscript, that the alpha finding and alpha/gamma coupling, 1/f etc... are all consistent with this strategy, but that's a discovery, not necessary a hypothesis that was initially conceived. If there was some rationale that I missed, I apologize.
2. The behavioral task would benefit from being explained better up front. From the text itself, it is hard to understand exactly what was done. I'm assuming that the cueing period includes either 1, 2, 3 or 4 simultaneous cues to indicate which features of the upcoming stimulus are relevant? If so, this needs to be spelled out as such. It might also be helpful to indicate this in Figure 1 as the current illustration has simply one of the above 4 possibilities (if my understanding is correct).
3. Related to the point above, the behavior of individuals needs to be plotted more fully/transparently. I can see that there is an attempt to do this in Fig. S1A1 but I don't think this is sufficient. Example of individual psychometric functions and group averages would be important to understand why, for example, color discrimination takes a bigger hit than direction going from a context of 1 to 4 targets. As far as I understand, these details are not accounted for in subsequent analysis as condition 1target vs. 4target are averaged regardless of what the relevant feature actually was.
4. The alpha finding is quite interesting, but again, it would be helpful to see clear time series of its phenomenology with respect to the phases of the task. Something similar to Fig. 4A that is in the context of cueing and stimulus presentation would be helpful. Even better, examples of a single subject under different load conditions would be great. The more raw data presented the better one can get an intuition of what is being reflected in summary statistics.
5. The thalamic findings are quite intriguing. A few years ago, Kimora et al. (Nature Neuroscience) saw the opposite type of activity in the macaque Pulvinar (a confidence signal). Here, the authors seem to describe an uncertainty signal in the MD (which is also observed in rodents). Some discussion on this (contrast with pulvinar) and its potential functional implications would be good.

In sum: I think this is excellent work and my comments are meant to constructively help the authors sharpen their message. It took me a couple of reads to go through this manuscript to appreciate it, but the general reader has to be able to do that on a single go.

Reviewer #2 (Remarks to the Author):

Signed, Peter Murphy

This manuscript by Kosciessa and colleagues provides new insights into ways in which uncertainty over feature relevance during perception of visual stimuli affects subsequent decisions about those stimuli. The authors developed a novel extension of an established perceptual decision-making paradigm (the random dot motion task) that permits parametric control of target feature load (uncertainty). They employ drift diffusion modelling constrained by non-invasive electrophysiology to isolate the specific decision computations that are affected by load-related uncertainty. They then leverage these modelling results to systematically interrogate – via multiple analysis approaches and measurement modalities – possible neural mechanisms facilitating adaptation to different levels of load-based uncertainty. They show that stimulus processing is associated with well-known frequency-specific responses in scalp EEG, which are in turn modulated by uncertainty and associated with individual differences in model-estimated uncertainty effects on decision-making behaviour. They show that putative metrics of neural ‘excitability’ (EEG signal entropy and power spectrum slope), as well as fast stimulus-related changes in arousal (read out via pupil diameter rate of change), are boosted under heightened uncertainty. Finally, the authors use functional MRI in the same cohort to show that uncertainty-driven increases in thalamic activity are related to this constellation of other effects.

The paper is truly impressive in its scope, technical sophistication, and rigour. It combines a clever and elegant extension of a classic perceptual decision-making paradigm with state-of-the-art approaches in human cognitive neuroscience. The alignment of multiple measurement modalities (psychophysics; EEG; fMRI; pupillometry) with cutting edge analysis methods (neurally constrained algorithmic modelling; advanced multivariate spectral and BOLD signal analysis) is extremely rare in the field and a pleasure to see together in a single manuscript. The results are also exciting and shed important new light on how the human brain flexibly adapts to make decisions in the face of varying contextual demands. The implication of the thalamus in particular is highly intriguing, and timely in that it adds to a growing literature, thus far predominantly from animal models and computational studies, on the role of this brain region in contextually-appropriate regulation of decision formation.

I think that the manuscript and supplement in their present forms are packed with interesting findings and almost publication ready. Below I outline some moderate to minor issues, as well as some questions that arose for me in my reading of the manuscript that the authors may wish to address through further analysis/discussion, or rather opt to keep for a follow-up paper on their rich dataset.

-- I think more could be done to clarify the nature of the task early on in the manuscript. In the final paragraph of the introduction, and in Figure 1B, the task is briefly explained but without explicit mention of the fact that it is an interrogation paradigm whereby participants only make their decisions after seeing the full multi-attribute stimulus for an experimenter-determined duration, once they are cued to

respond by the probe. As a consequence, it is also unclear for much of the early part of the manuscript that the drift diffusion modelling is being applied specifically to probe-aligned behaviour, and that the putative decision process does not receive input during stimulus presentation, but rather only upon probe onset (presumably via a working/eidetic memory trace; see my next point). Some information about the timing of the task in the main text (in particular that the multi-attribute stimulus is viewed for a fixed 3 s) would also be welcome. Clarifying these aspects would in turn render somewhat counterintuitive statements in the final Introduction paragraph – specifically, that ‘decreased accumulation’ was found to be associated with *increased* excitation – more understandable, since it is then clear that the increased excitation applies specifically to the pre-decisional ‘stimulus encoding’ period.

-- One outstanding question that remains unanswered in the manuscript in its present form is how exactly processing of the multi-attribute stimulus during its presentation is translated into the post-probe decisions that are the subject of the algorithmic modelling. The modelling seems to make the implicit assumption that the stimulus viewing period is entirely devoted to stimulus encoding, and actual decision formation only begins after probe onset. But of course this need not be the case: one alternative scenario is that decisions about each target feature (as defined by the initial cue) are in fact made *during* stimulus presentation, and the post-probe period is solely dedicated to translating the outcomes of those internal decisions (maintained in working memory) into appropriate motor responses. This distinction matters, because it would constrain what kind of process (sensory encoding or actual decision formation) the various neural signatures identified by the authors (cortical excitability, arousal, thalamic activity) are regulating. The authors have at their disposal a means for arbitrating between these possibilities. Specifically, interrogating pre-probe motor preparation signals for trials on which only a single feature is cued, or multiple features that all converge on the same motor response. If choice-selective motor preparation is observed during the stimulus viewing period, this would strongly suggest that decision formation is taking place before probe onset. Note that I am not convinced that the absence of a clear CPP effect during the stimulus period (Figure S2B) rules out pre-probe decision formation: we don’t have a strong expectation for how the stimulus-aligned CPP should behave in this kind of multi-attribute, high feature-uncertainty, delayed-response task, and in any case the 0.5 Hz high-pass filtering applied to this signal effectively rules out observing the kind of slow ramping that may accompany decision formation during this (3 s) period.

-- There is currently no illustration of the quality of the DDM model fits. I would strongly encourage the authors to include an extra supplement (or extra panels in Figure S1) that visually illustrate qualitative goodness of fit – for example by plotting observed/model-predicted RT distributions for each level of target load.

-- The gamma frequency band seems to be defined differently for different analyses (i.e. 40-90 Hz in Fig. 3; 60-150 Hz in Fig. 4), and indeed the effects in Fig. 4 seem to lie primarily outside of the band focussed on in Fig. 3. Ideally these would be consistent. Is there a reason why the Fig. 3 analysis did not extend higher than 90 Hz. Indeed, the brainscores seem to be highest toward the upper bound of the examined range (Fig. 3D), suggesting some meaningful signal may lie above the upper bound of that reported and may indeed reconcile the results in Figs. 3 and 4.

-- The distinction between the ‘task PLS’ and the ‘behavioural PLS’ is not very clear in the main text – in particular, it is introduced early on with passing reference to supplementary materials (line 199), but

then becomes quite important in light of the various findings implicating specifically thalamic BOLD responses in adjustments to task load (Fig. 7). Indeed, I find even the additional explanation given in text S4 to be quite unclear and it makes reference to supplementary figure panels (Fig. S4A/B) that I do not know how to interpret given the scant information in the associated legend. In sum then, some additional clarification in the main text about what differentiates task PLS from behavioural PLS and how each should be interpreted is required, early on where they are first introduced.

Minor issues:

-- Intro, line 35: "parallel" here seems misleading since EEG and fMRI were measured in separate sessions

-- Intro, line 47: "broad sensitivity increases". What is meant by this term? Sensitivity in the decision-making literature is a loaded term, typically referring to an increase in perceptual sensitivity captured by measures like d' . Perhaps consider rephrasing.

-- Line 101: typically the acronym "CPP" refers to centro-parietal positivity, not centroparietal positive potential (cf. the O'Connell, Kelly papers where this component was first described).

-- Line 147: please define "response agreement". I think the term "response convergence" is also used at various other parts of the manuscript – please pick one and use consistently throughout.

-- Fig S2A: is y-axis of inset meant to be CPP slope, or μ/β slope? Also, legend for panel A refers to subpanels A/B which I think don't exist.

-- Fig S3: there seems to be some confusion of panel labels between the figure and the figure legend. Also, there is not a consistent description of what the arrows reflect in panels B and C.

-- The phrase "selective attention" or "selective attention condition" is used at several points in the main text (e.g. p.9) without explicit description of what this actually means. I assume it is meant to refer to the single-target condition, where participants can selectively attend to a single feature. But this is not clear, and also I think not that helpful (one could argue that all four levels of load vary in the 'degree' of selective attention that they permit, so I don't think it makes much sense to designate only one level as the 'selective attention condition'). I suggest sticking closer to the actual design and simply refer to this as the single-target condition; further clarification about the high level of selective attention that this condition, relative to the others, permits can then be delivered as desired.

-- Fig. 3A: references to "theta" should rather be "delta/theta", since the frequency band in question (2-8 Hz) covers both delta (usually ~1-3 Hz) and theta (usually ~4-8 Hz) bands.

-- Fig. S4: there's a lot going on in this figure and the legend is extremely brief. The legend entries for several individual panels (in particular A-C) need considerable elaboration.

-- Lines 242-244: "...the magnitude of individual entropy modulation in this cluster scaled with increases in the SPMF, indicating that alpha desynchronization was accompanied by broadband changes in signal irregularity". From what I understand the SPMF is a composite multivariate measure that incorporates relationships between frequency-specific power changes and target load (uncertainty) across *all* considered frequencies. So it is non-specific in the sense that it does not parse theta-band from alpha-band contributions. How, then, can an entropy/SPMF correlation specifically support the inference of an alpha/entropy relationship?

-- Line 296: "fast integrators" is not very clear. I suggest replacing with "individuals with higher drift rates". After all, drift rate captures more than just integration speed – it's really a measure of the

relative signal-to-noise (assuming a fixed within-trial noise parameter) of the evidence source.

- Line 299: "This suggests that arousal...related to increases in local cortical excitability..." This seems too strong given that the only observed relationship is between pupil and SPMF, and *not* with putatively more specific measures of excitability (entropy and power spectrum slope). On a related note, was there any frequency-specificity to the pupil effect, or did pupil correlate with the range of spectral modulations that combine to make up the SPMF effect?
- Lines 649-650: RGB values seem to be incorrect. Specifically, the value for 'low saturation green' will produce a red colour, while 'high saturation red' will produce a green colour.
- P. 31-32: why was baselining applied to gamma-band estimates but not lower frequencies?
- Line 817: reference to Figure 1A seems to be incorrect.

Reviewer #3 (Remarks to the Author):

Thank you for inviting me to review this manuscript by Kosciessa and colleagues, in which the authors conducted a series of computational neuroimaging studies in order to better understand how the brain deals with uncertainty.

The authors hypothesized that uncertainty would facilitate a shift from selective to diffuse, asynchronous processing, which they further suggest should align with activity in the the thalamus. To test these hypotheses, the authors had participants perform a parametric adaptation of the classic dot motion task, in which they manipulated the number of stimulus dimensions that were task-relevant in a given trial while holding the sensory features of the task (i.e., its appearance on the screen) constant across trials. Probe uncertainty was parametrically manipulated using valid pre-stimulus cues, indicating the feature set from which a probe would be selected.

The authors then use an impressive range of different computational and statistical approaches to conclude that task uncertainty is related to lowered drift rate, asynchronous EEG signatures, dilated pupils and upregulated activity in the thalamus. They conclude that neuromodulatory processes involving the thalamus shape cortical excitability states in humans, and that a shift from alpha-rhythmic to aperiodic neural dynamics adjusts the processing fidelity of external stimuli in service of upcoming decisions.

Overall, I found the manuscript to be clear, well-motivated and professionally executed. I have only minor questions for the authors that I hope will help to clarify the main results.

I didn't quite understand the rationale for fixing the threshold across target load levels in the model. Surely the threshold was found to change loads? If so, is it not an important feature of any resultant model? Do the authors results hold if the measure is not held constant?

Drift rate looks like it shows a non-linear effect of parametric uncertainty, whereas NDT smoothly increased as a function of load. Do the authors have an explanation for why the two measures showed different interactions with parametric uncertainty?

I'm sorry if I missed this, but how stable were the results from the hDDM model? What kind of model search did the authors use so as to ensure stability?

The thalamic loadings in the first latent component of the PLS in Figure 7D1 are reminiscent of Ted Jones' "matrix" regions (Jones, 2001). The authors may wish to comment on this potential overlap, which can be quantified using the calcium binding protein Calbindin (Niemann et al., 2000; Müller et al., 2020).

Dear Reviewers,

thank you for your encouraging and constructive comments. We appreciate your input on the previous version of our manuscript and hope to have satisfactorily addressed your questions, remarks and suggestions with this revision. We highlighted changes to the materials in **yellow**.

In particular, we now (a) provide a clearer and more comprehensive task description early on in the paper, (b) clarify our hypotheses regarding alpha and entropy, (c) discuss the results regarding the thalamus more extensively. We also implemented more specific requests, such as additional analyses on feature-specific behavior, pre-probe motor preparation, and qualitative goodness-of-fit assessments of the drift diffusion model. If needed, we report additional data in our replies.

We would like to note that we intend to make the following questions and responses available alongside the manuscript as part of Nature Communication's transparent peer review system (in the case of acceptance). As such, we intend for our responses in this letter to also be available to the interested reader, also when we opted to not include additional analyses in the manuscript or its main supplement.

Reviewer #1:

The manuscript by Kosciessa et al utilizes a combination of EEG, fMRI and a perceptual decision making task to identify shifts in cortical activity patterns that underlie perceptual uncertainty on one end and the involvement of the thalamus in potentially mediating these shifts, on another. The study is quite topical and of broad relevance. I really enjoyed reading the manuscript and very much appreciate the rigor and thoroughness here.

We thank the reviewer for this encouraging review.

Below are a few points of clarification that should help make the manuscript more accessible and easier to read:

1. The premise of the author's hypothesis that heightened uncertainty would shift cortical state is never spelled out. I recognize that, after going through the manuscript, that the alpha finding and alpha/gamma coupling, 1/f etc... are all consistent with this strategy, but that's a discovery, not necessary a hypothesis that was initially conceived. If there was some rationale that I missed, I apologize.

We thank the reviewer for indicating that this hypothesis should be spelled out more clearly. We have edited the second paragraph (page 3f.) to clarify this hypothesis (text shown below). Furthermore, we have restructured Figure 1 and added a new subpanel to (reproduced below) to illustrate our motivation regarding alpha/gamma coupling, entropy and 1/f hypotheses. We hope that this better introduces our hypothesis concerning cortical state shifts from the beginning.

*"We hypothesize that such uncertainty-related processing adjustments involve a switch between different cortical states (Figure 1B). Specifically, selective gain control has been associated with **phasic** (i.e., phase-dependent) inhibition of task-irrelevant stimulus dimensions during cortical alpha (~8-12 Hz) rhythms (Klimesch, Sauseng, & Hanslmayr, 2007). Conceptually, such rhythmic modulations of feedforward excitability (Haegens, Nacher, Luna, Romo, & Jensen, 2011; Lorincz, Kekesi, Juhasz, Crunelli, & Hughes, 2009) provide temporal 'windows of opportunity' for high-frequency gamma synchronization in sensory cortex (Spaak, Bonnefond, Maier, Leopold, & Jensen, 2012) and increased sensory gain (Fries, 2015). However, specifically increasing the fidelity of single stimulus dimensions is theoretically insufficient when uncertain environments require joint sensitivity to multiple stimulus features (Mo et al., 2019; Pettine, Louie, Murray, & Wang, 2020). Alternatively, transient increases to the **tonic** excitation/inhibition (E/I) ratio in sensory cortex provide a principled mechanism for such elevated sensitivity to – and a more faithful processing of – high-dimensional stimuli (Destexhe, Rudolph, & Pare, 2003). In electrophysiological recordings, scale-free 1/f slopes are sensitive to differences in E/I ratio (Gao, Peterson, & Voytek, 2017) and vary alongside sensory stimulation (Podvalny et al., 2015). Relatedly, sample entropy provides an information-theoretic index of signal irregularity that is highly sensitive to scale-free content and may thus similarly track excitability (Kosciessa, Kloosterman, & Garrett, 2020). However, whether contextual demands modulate scale-free activity and/or entropy is unknown. We hypothesize that heightened uncertainty shifts cortical states from a regime of rhythmic excitability modulations in the alpha band (associated with a modulation of gamma-band activity) towards tonic excitability increases (as indexed via increased scale-free irregularity and neural entropy; Figure 1B)."*

Figure 1B. Hypothesized phasic and tonic excitability modulation in cortex as a function of uncertainty. Rhythmic (alpha) fluctuations in EEG are thought to reflect alternating phases of relative excitatory and inhibitory dominance (Atallah & Scanziani, 2009; Klimesch et al., 2007; Poo & Isaacson, 2009), yielding variations in high-frequency (gamma) power as a function of low-frequency phase. In contrast, static increases in excitatory-to-inhibitory tone may alter aperiodic signal dynamics, reflected in a flattening of spectral slopes in the frequency domain (Gao et al., 2017) and relative increases in sample entropy as an index of signal irregularity.

2. The behavioral task would benefit from being explained better up front. From the text itself, it is hard to understand exactly what was done. I'm assuming that the cueing period includes either 1, 2, 3 or 4 simultaneous cues to indicate which features of the upcoming stimulus are relevant? If so, this needs to be spelled out as such. It might also be helpful to indicate this in Figure 1 as the current illustration has simply one of the above 4 possibilities (if my understanding is correct).

We thank both Reviewers 1 and 2 for indicating problems with the initial task description. We clarified in Figure 1D (reproduced below) and its caption that the cueing period indeed included "1, 2, 3 or 4 simultaneous cues to indicate which features of the upcoming stimulus are relevant". We now highlight this by showing only two of the four cues on the uniform black background, while all potentially available cues are shown on a separate grey background.

Figure 1D. Participants performed a *Multi-Attribute Attention Task* ("MAAT") during which they had to sample up to four visual features in a joint display for immediate subsequent recall. On each trial, participants were first validly cued to a target probe set (here direction and size). The stimulus (which always contained all four features) was then presented for 3 sec, and was followed by a probe of one of the cued features (here, which direction was most prevalent in the stimulus). The number and identity of simultaneous pre-stimulus cues were fixed for blocks of 8 trials to experimentally manipulate the level of expected (ongoing) probe uncertainty.

We report further changes to the task description in the response to Reviewer 2's Q1 below.

3. Related to the point above, the behavior of individuals needs to be plotted more fully/transparently. I can see that there is an attempt to do this in Fig. S1A1 but I don't think this is sufficient. Example of individual psychometric functions and group averages would be important to understand why, for example, color discrimination takes a bigger hit than direction going from a context of 1 to 4 targets. As far as I understand, these details are not accounted for in subsequent analysis as condition 1target vs. 4target are averaged regardless of what the relevant feature actually was.

The reviewer is correct in stating that the study provides no information regarding psychometric functions. Due to the use of fixed parameters for individual features (e.g., the

same motion coherence) across subjects and conditions, we cannot perform a full psychometric analysis. As such, we are unfortunately unable to elucidate why “color discrimination takes a bigger hit than direction going from a context of 1 to 4 targets”. The reviewer correctly assumes that these differences – insofar as they exist – are not accounted for in the contrast of interest. However, this is by design, as the task targets assumedly feature-invariant properties. In particular, (A) by keeping the features constant across target loads, we prevent “bottom-up” differences in stimulus-driven features. Given that all feature conjunctions are within-subject balanced across target load conditions, we find our averaging strategy to be straightforward without assuming a major bias arising from individual features. Note also that although the low-level feature differences *could* vary somewhat between sessions (and thus the setting and screen restrictions in the MR assessment), we interpret the observed reliability between sessions as a positive sign that low-level stimulus features do not significantly contribute to our results of interest. Further, (B) our supplementary analysis summarized in Figure S2D targeted the question alluded to by the reviewer, namely whether the effect of preceding target load on evidence drift (indicated via CPP) was dependent on the specific feature that was probed. In line with our suggestion of a rather feature-invariant effect, a linear decrease of CPP slope was indicated regardless of the specific feature that was probed.

Extending this analysis to ‘raw’ behavioral markers, we similarly observed a main effect of target load on RT and accuracy for all probed stimulus features other than direction (see Figure R1A). However, the reviewer correctly notes that the effect of preceding target load on behavioral discrimination accuracy and speed (*indirectly* assessed via the probe) varies as a function of the probed feature (see Figure R1B). The lack of accuracy (and the least RT) modulation for direction probes is interesting, given that evidence integration as indicated by CPP slopes was highest for direction probes (see Figure S2C2). Whether this highlights problems with the coherence setting or designates a special role for motion detection remains unclear from the current results and would require a more stringent psychophysical analysis in future work as the reviewer suggests.

In any case, it is crucial to note that the effect size on average log RTs of the interaction between (a) probed feature and target amount is small (EEG: partial $\eta^2 = .07$; MRI: partial $\eta^2 = .03$) compared to the substantial main effect of (b) target amount (EEG: partial $\eta^2 = .83$, MRI: partial $\eta^2 = .80$). Similarly, weak feature x target amount interaction effects hold for accuracy (a; EEG: partial $\eta^2 = .03$ MRI: partial $\eta^2 = .01$) compared to the main effect for target amount alone (b; EEG: partial $\eta^2 = .09$, MRI: partial $\eta^2 = .05$). [Note that effect sizes were estimated by comparing mixed effects models that included the additional dependent variable under investigation to a more constrained model; in the case of (a) adding the interaction term to a model including both main effects of probed feature and target amount, and in the case of (b), adding the main effect of target amount to a null model only including the main effect of probed feature.] In sum, these results suggest that the analyses in the manuscript – as intended – describe relatively feature-independent uncertainty effects that we targeted in our design.

Figure R1. Linear effect of target load on mean log-RT and accuracy as a function of probed feature. Red indices indicate results from the EEG session, grey indices indicate results from the MRI session. Asterisks indicate significant differences from zero as assessed via paired t-tests (** $p < .05$). (B) Differences in linear target load effects by feature. Linear beta coefficients have been averaged across the EEG and fMRI sessions. Only significant pairwise comparisons are shown (** $p < .01$, *** $p < .001$).

4. The alpha finding is quite interesting, but again, it would be helpful to see clear time series of its phenomenology with respect to the phases of the task. Something similar to Fig. 4A that is in the context of cueing and stimulus presentation would be helpful. Even better, examples of a single subject under different load conditions would be great. The more raw data presented the better one can get an intuition of what is being reflected in summary statistics.

We appreciate and share the reviewer's interest in single-trial dynamics of alpha phenomenology. Properly presenting such phenomenology is not straightforward, as it requires high single-trial SNR and is subject to selection effects. In Figure R2, we show a time series from a single subject with high accuracy performance from two different target conditions to exemplify the phenomenology. There are notable questions regarding the representativeness of such an assessment, including trial-by-trial variability (neural as well as potentially non-neural), the lack of a depiction of (substantial) inter-individual differences, and the possibility of selection biases such as time on task. For this reason, we currently do not highlight such examples in the manuscript. Nonetheless, we hope that the reviewer will agree that the examples provided below provide single-trial, single-person illustrations of the effects observed the aggregate group level, such as increases in pre-stimulus alpha power with the number of targets as well as a reduction in alpha (power and duration) during the stimulus presentation period. We would be happy to include this figure in the manuscript if deemed useful by the reviewer.

Figure R2. Exemplary broadband time series averaged across occipital channels (O1, Oz, O2). Grey shading indicates the period of cue presentation, yellow shading indicates stimulus presentation. Examples are from the subject with the highest accuracy in the single-target condition.

5. The thalamic findings are quite intriguing. A few years ago, Kimora et al. (Nature Neuroscience) saw the opposite type of activity in the macaque Pulvinar (a confidence signal). Here, the authors seem to describe an uncertainty signal in the MD (which is also observed in rodents). Some discussion on this (contrast with pulvinar) and its potential functional implications would be good.

We now provide an extended discussion of thalamic results in the main text (p. 20f.; reproduced below in quotes), including a reflection on the potential differences between MD and Pulvinar signals. We would like to note that while a possible dissociation of MD and Pulvinar is indeed interesting, it is likely that both regions jointly contribute to task processing, especially when they involve visual attention as in the MAAT (Halassa & Kastner, 2017).

Unfortunately, we were unable to find specific references to an ‘uncertainty signal’ in the MD in rodents. To our knowledge, the most closely related studies to the attentional manipulation in our task assessed MD activity during selective attention cues in rodents (e.g., Rikhye, Gilra, & Halassa, 2018; Schmitt et al., 2017), but did not assess a condition in which both conditions are simultaneously cued (which would align with the target uncertainty manipulation here). Intriguingly, this line of research indicates a representation of the cue context in MD, which supports the selection of relevant rules in PFC (Rikhye, Gilra, et al., 2018). We now note this in the discussion. Moreover, we are aware of similar uncertainty-related statements (based on macaque studies) such as that “the role of the MD may be most relevant when the PFC is multitasking for long time intervals” (Pergola et al., 2018, p. 1021). Related arguments have also been made in the domain of reward updating, where MD presumably “allows for the rapid discovery and persistence with rewarding options, particularly in uncertain or changing environments” (Chakraborty, Kolling, Walton, & Mitchell, 2016, p. 1). We apologize if we missed any other specific papers and would be happy to include them in our discussion if the reviewer would point us to them.

“In particular, anterior and midline thalamic nuclei (such as the mediodorsal nucleus (MD), in which neuro-behavioral relations were maximal in our results) are implicated in establishing (Rikhye, Gilra, et al., 2018), sustaining (Bolkan et al., 2018), and switching (Marton, Seifkar, Luongo, Lee, & Sohal, 2018; Parnaudeau et al., 2013; Rikhye, Gilra, et al., 2018; Wright, Vann, Aggleton, & Nelson, 2015) prefrontal rule representations depending on the active task context (Rikhye, Gilra, et al., 2018). An intriguing possibility is that target uncertainty increases MD engagement to enable a more dynamic target selection among high-dimensional prefrontal feature representations (Mack, Preston, & Love, 2020; Rigotti et al., 2013; Rikhye, Gilra, et

al., 2018). Such proposal aligns with highest MD engagement during the integration of multiple cognitive demands (Mitchell, 2015; Pergola et al., 2018), and MD lesions specifically impairing performance for larger stimulus sets (Parker, Eacott, & Gaffan, 1997) and more complex tasks (Edelstyn, Mayes, & Ellis, 2014). As such, MD may play a crucial role in flexible behaviors, particularly under uncertainty (Chakraborty et al., 2016; Mitchell, 2015; Pergola et al., 2018, p. 1017; Soltani & Izquierdo, 2019).

Such uncertainty modulation in anterior-medial thalamic regions complements the previously reported representation of perceptual precision or confidence in Pulvinar neurons (Jaramillo, Mejias, & Wang, 2019; Komura, Nikkuni, Hirashima, Uetake, & Miyamoto, 2013). Such 'perceptual priors' may coordinate feature-selective information within and across visual cortex (Halassa & Kastner, 2017; Jaramillo et al., 2019; Saalmann, Pinsk, Wang, Li, & Kastner, 2012) once perceptual targets have been selected. Notably, the MAAT manipulates probe uncertainty, but not the sensory information (e.g., motion coherence) per se. It is thus an intriguing question whether a complementary manipulation of visual characteristics could dissociate MD and Pulvinar engagement in future work."

In sum: I think this is excellent work and my comments are meant to constructively help the authors sharpen their message. It took me a couple of reads to go through this manuscript to appreciate it, but the general reader has to be able to do that on a single go.

We again wish to thank the reviewer for their time in reviewing our submission and hope that our revision helps to convey our main messages also to a broader readership.

Reviewer #2:

Signed, Peter Murphy

This manuscript by Kosciessa and colleagues provides new insights into ways in which uncertainty over feature relevance during perception of visual stimuli affects subsequent decisions about those stimuli. The authors developed a novel extension of an established perceptual decision-making paradigm (the random dot motion task) that permits parametric control of target feature load (uncertainty). They employ drift diffusion modelling constrained by non-invasive electrophysiology to isolate the specific decision computations that are affected by load-related uncertainty. They then leverage these modelling results to systematically interrogate – via multiple analysis approaches and measurement modalities – possible neural mechanisms facilitating adaptation to different levels of load-based uncertainty. They show that stimulus processing is associated with well-known frequency-specific responses in scalp EEG, which are in turn modulated by uncertainty and associated with

individual differences in model-estimated uncertainty effects on decision-making behaviour. They show that putative metrics of neural ‘excitability’ (EEG signal entropy and power spectrum slope), as well as fast stimulus-related changes in arousal (read out via pupil diameter rate of change), are boosted under heightened uncertainty. Finally, the authors use functional MRI in the same cohort to show that uncertainty-driven increases in thalamic activity are related to this constellation of other effects.

The paper is truly impressive in its scope, technical sophistication, and rigour. It combines a clever and elegant extension of a classic perceptual decision-making paradigm with state-of-the-art approaches in human cognitive neuroscience. The alignment of multiple measurement modalities (psychophysics; EEG; fMRI; pupillometry) with cutting edge analysis methods (neurally constrained algorithmic modelling; advanced multivariate spectral and BOLD signal analysis) is extremely rare in the field and a pleasure to see together in a single manuscript. The results are also exciting and shed important new light on how the human brain flexibly adapts to make decisions in the face of varying contextual demands. The implication of the thalamus in particular is highly intriguing, and timely in that it adds to a growing literature, thus far predominantly from animal models and computational studies, on the role of this brain region in contextually-appropriate regulation of decision formation.

I think that the manuscript and supplement in their present forms are packed with interesting findings and almost publication ready. Below I outline some moderate to minor issues, as well as some questions that arose for me in my reading of the manuscript that the authors may wish to address through further analysis/discussion, or rather opt to keep for a follow-up paper on their rich dataset.

We thank the reviewer for this encouraging review and the following constructive comments. We hope we clarified the open issues in the current revision, and in the responses below. We also would like to thank the reviewer for the helpful list of detailed minor comments of smaller inconsistencies or errors that we missed in the previous submission.

1. I think more could be done to clarify the nature of the task early on in the manuscript. In the final paragraph of the introduction, and in Figure 1B, the task is briefly explained but without explicit mention of the fact that it is an interrogation paradigm whereby participants only make their decisions after seeing the full multi-attribute stimulus for an experimenter-determined duration, once they are cued to respond by the probe. As a consequence, it is also unclear for much of the early part of the manuscript that the drift diffusion modelling is being applied specifically to probe-aligned behaviour, and that the putative decision process does not receive input during stimulus presentation, but rather only upon probe onset (presumably via

a working/eidetic memory trace; see my next point). Some information about the timing of the task in the main text (in particular that the multi-attribute stimulus is viewed for a fixed 3 s) would also be welcome. Clarifying these aspects would in turn render somewhat counterintuitive statements in the final Introduction paragraph – specifically, that ‘decreased accumulation’ was found to be associated with *increased* excitation – more understandable, since it is then clear that the increased excitation applies specifically to the pre-decisional ‘stimulus encoding’ period.

We added detail to the final part of the introduction (page 3f.; text repeated below in quotations), and the figure caption of Figure 1D to clarify the points brought up by the reviewer, including that (a) drift diffusion modelling is applied specifically to probe-aligned behaviour, (b) DDM results likely pertain to working/eidetic memory traces, not sensory processing, (c) increased excitation applies specifically to the pre-decisional ‘stimulus encoding’ period. We also added information regarding the task timing to Figure 1D and the main text (e.g., p. 5 : “Multi-attribute stimuli were shown for a fixed duration of three seconds, after which subjects were probed as to which of the two exemplars of a single feature was most prevalent (via 2-AFC).”).

“Here, we aimed at overcoming this lacuna by assessing the effects of contextual uncertainty during stimulus encoding on cortical excitability, neuromodulation, and thalamic activity in humans. We performed a multi-modal EEG-fMRI experiment, measuring the same participants in two separate sessions, to capture both fast cortical dynamics (EEG) and subcortical activity (fMRI), while recording pupil dilation as a non-invasive proxy for neuromodulatory drive (Reimer et al., 2014). Participants performed a parametric adaptation of the classic dot motion task (Gold & Shadlen, 2007), in which individual stimulus elements were defined by a conjunction of color, size, direction and luminance features (Figure 1D). By presenting valid cues (1-4 cues shown simultaneously) in advance of stimulus presentation, we manipulated the number of stimulus dimensions that are task-relevant in a given trial, while holding the sensory features of the task (i.e., its appearance on the screen) constant. Following the presentation of the multi-attribute stimuli, participants were probed with respect to a single target feature that was selected from the cued set. By applying drift-diffusion modeling to participants’ probe-related choice behavior while jointly assessing electrophysiological signatures of corresponding decision processes, we found that uncertainty during sensation reduced the rate of subsequent evidence integration, suggesting reductions in the precision of encoded target information. This uncertainty-related reduction in available evidence following the probe was associated with increased cortical excitability during the (pre-decisional) stimulus presentation period, as indexed by joint low-frequency (~alpha) desynchronization and high-frequency (~gamma) synchronization, and an increase in E/I ratio (as indicated by increased sample entropy and flatter scale-free 1/f slopes). These excitability adjustments, potentially reflecting a joint encoding of multiple features during periods of higher uncertainty, occurred in parallel with increases in pupil-based arousal.”

2. One outstanding question that remains unanswered in the manuscript in its present form is how exactly processing of the multi-attribute stimulus during its presentation is translated into the post-probe decisions that are the subject of the algorithmic modelling. The modelling seems to make the implicit assumption that the stimulus viewing period is entirely devoted to stimulus encoding, and actual decision formation only begins after probe onset. But of course this need not be the case: one alternative scenario is that decisions about each target feature (as defined by the initial cue) are in fact made *during* stimulus presentation, and the post-probe period is solely dedicated to translating the outcomes of those internal decisions (maintained in working memory) into appropriate motor responses. This distinction matters, because it would constrain what kind of process (sensory encoding or actual decision formation) the various neural signatures identified by the authors (cortical excitability, arousal, thalamic activity) are regulating. The authors have at their disposal a means for arbitrating

between these possibilities. Specifically, interrogating pre-probe motor preparation signals for trials on which only a single feature is cued, or multiple features that all converge on the same motor response. If choice-selective motor preparation is observed during the stimulus viewing period, this would strongly suggest that decision formation is taking place before probe onset. Note that I am not convinced that the absence of a clear CPP effect during the stimulus period (Figure S2B) rules out pre-probe decision formation: we don't have a strong expectation for how the stimulus-aligned CPP should behave in this kind of multi-attribute, high feature-uncertainty, delayed-response task, and in any case the 0.5 Hz high-pass filtering applied to this signal effectively rules out observing the kind of slow ramping that may accompany decision formation during this (3 s) period.

We agree with the observation that some uncertainty remains regarding the actual process that is regulated in this paradigm, although we would also like to note that dissociating encoding and decision formation is generally difficult in other paradigms commonly used in the field. As hinted at, recording participants' decisions via post-stimulus probes renders fine-grained insights into constituent processes of behavioral outcomes difficult.

The reviewer notes that "The modelling seems to make the implicit assumption that the stimulus viewing period is entirely devoted to stimulus encoding, and actual decision formation only begins after probe onset." Notably, the DDM model allows for differences in starting point of the accumulation process, which would conceptually correspond to the availability of a 'pre-planned' response. Including in our model a starting point parameter that varied by target load did not fit substantially better than a model without such a parameter as judged by DIC, and results for the drift rate changes of interest did not qualitatively differ in this model either. We would also like to note that we performed a supplementary HDDM analysis targeting the question whether condition differences in response convergence (i.e., when cued features converge on the same left or right response for their more prevalent options during the trial) could explain the target condition differences. In this analysis, despite controlling for response convergence (and thus the ability to pre-plan motor responses), we still observed highly similar results concerning drift rate modulation as a function of target number. See the next section below for more details.

To more directly address the reviewer's question, we interrogated pre-probe motor preparation signals for trials on which only a single feature is cued, or multiple features that jointly converge on the same motor response (i.e., where the winning options for all features would require the identical left/right response). Results are shown in Figure R3. This analysis indicated no differences in contralateral mu-beta power (which was the analysis target in Figure S2A, and for which motor preparation is typically associated with a power reduction, signifying disinhibition) immediately prior to probe presentation for the single-target condition, but suggested elevated mu-beta power pre-probe ipsilaterally for single targets, potentially in line with stronger inhibition of the alternative motor option already during stimulus presentation. Given that mu-beta power by itself does not provide sufficient evidence for representations of the chosen option, we performed an additional decoding analysis that we also now include in the Supplement (Text S2, Figure S2C, reproduced below). In short, this analysis suggests that some information regarding the subsequently executed motor laterality can be decoded (from mu-beta power and broadband amplitudes) just prior to probe presentation in the single-target condition.

Figure R3. Interrogating pre-probe motor preparation signals. (Top) Contralateral mu-beta power does not significantly differ between conditions prior to probe onset. (Bottom) Ipsilateral (see Figure S2A topography inset) mu-beta power diverges between conditions prior to probe-onset, with larger power (indicating stronger inhibition of the non-chosen motor response) when motor responses could be pre-planned in the single-target condition. Red lines below the plot indicate statistically significant differences between single and multi-target conditions with no convergence as indicated via paired t-tests (cluster-definition threshold $p < .05$, corrected significance level $p < .05$). Orange lines indicate differences between multi-target conditions with and without convergence, respectively. Shaded areas indicate within-subject standard errors.

Text S2. Decoding motor preparation signals. We performed a decoding analysis to further explore the emergence of response-specific information across the trial. In particular, we trained a decoder based jointly on the spatial topography of the single-trial broadband time series (~ single-trial CPP), and of the average 8-15 Hz power (estimated using 7 cycle wavelets), both response-locked. Broadband and mu-beta signals were spatially concatenated at each time point, following z-scoring across channels within each measure. This response-locked motor execution classifier was then tested on each time point of signals aligned with stimulus onset (and thus probe onset by virtue of the fixed 3s stimulus presentation period) to assess the ability to classify motor preparation prior to probe onset. For classification analyses, we used linear support-vector machines (SVM) (K. R. Muller, Mika, Ratsch, Tsuda, & Scholkopf, 2001) via the libsvm implementation (www.csie.ntu.edu.tw/~cjlin/libsvm). In particular, we decoded ipsi- vs. contralateral response execution in each trial (in line with the lateralized mu-beta analyses). The analysis was separately performed for four splits of trials: (1) all trials, (2) single-target condition trials, (3) multi-target condition trials, in which all cued features converge on the same left or right response for their respective prevalent options during the trial (“response convergence”), (4) multi-target condition trials, in which at least one response would not converge with all other responses. Within each group, the minimum number of available trials across the left and right conditions were randomly selected and split into training (90% of trials) and test sets 100 times for cross-validation. [This

approach maximizes the available trials for each split, but includes unequal numbers of trials between splits. Hence, differences in decoding levels between splits should be interpreted with caution.] Results showed that classification accuracy for left vs. right responses was notably above chance and maximal at response, highlighting the adequacy of the response-aligned classifier (Figure S2C1). Applying this “decoder” to signals aligned to stimulus onset indicated (a) slightly above chance classification for single targets, and (b) slightly below chance classification for multi-targets without response convergence just prior to response (Figure S2C2). The latter may indicate a representation of the unobserved, alternate motor response, potentially suggesting “changes of mind” following probe presentation. These results jointly allude to the detectable presence of some decision information just prior to probe presentation (but not during a protracted period during stimulus presentation), while also highlighting that much of the eventual response information emerges following probe presentation.

Figure S2C. Motor response can only be decoded briefly prior to probe onset in 1-target condition. (C1) Decoding accuracy of executed motor response (left/right) in response-aligned signals. Maximum decoding accuracy was observed around the time of response for all assessed conditions. (C2) Decoding performance for stimulus-/probe-aligned signals. Decoders were trained on the response-aligned signals as in panel A. Grey shading indicates the stimulus presentation period. The offset of stimulus presentation coincides with probe onset. Note that due to the discrete periods, pre-probe above/below chance decoding is unlikely to have resulted exclusively from temporal smoothing due to the wavelet transform. For visualization, but not statistics, data in C2 were smoothed via moving mean averages of 10 adjacent sample points. Lines below the plot indicate statistically significant deviations from chance decoding (chance level of 50%, paired t-test, cluster-definition threshold $p < .05$, corrected significance level $p < .05$). Shaded bars indicate standard errors of the mean.

Taken together, these analyses suggest that partial decision formation may have occurred prior to probe presentation when afforded by the cue, which is also intuitive given the fixed button mapping. However, we do not agree that the onset of a decision process prior to probe presentation can definitively constrain which process is modulated. The presence of partial pre-probe decision formation does not suggest that this pre-probe decision formation is exclusively affected by our uncertainty manipulation. Regardless of whether decisions were partly (or in extreme scenarios even exclusively) formed during the encoding phase, we still observe a relation between neural adaptations during sensory encoding and effects on subsequent decision formation/execution/recall of prepared responses (given that individual drift rates correspond to CPP slopes estimated immediately prior to response).

3. There is currently no illustration of the quality of the DDM model fits. I would strongly encourage the authors to include an extra supplement (or extra panels in Figure S1) that visually illustrate qualitative goodness of fit – for example by plotting observed/model-predicted RT distributions for each level of target load.

We thank the reviewer for this suggestion and now provide a visual illustration of the qualitative goodness of fit. As suggested, we plot the observed and model-predicted RT distributions for target load levels one and four as an additional panel in Supplementary Figure S1G (pasted

below). As the reviewer likely appreciates, plotting individual plots would be difficult due to the large number of conditions*subjects in our design. Individual posterior predictive checks are however shown in our shared Jupyter notebooks*. We consider the qualitative fit appropriate, reflecting the increases of RT and decreases in accuracy as a function of target number. However, we also note that the fit is not perfect. In particular, the RT of incorrect responses appears to be slightly underestimated, potentially due to a comparatively low number of incorrect responses.

Figure S1G. Qualitative model fits in the selected HDDM model (exemplarily shown for the EEG session). Negative RTs correspond to wrong responses. Model-based (“posterior predictive”) values were sampled 50 times within each subject and condition (as implemented in the HDDM package), and probability density (100 RT bins) was estimated first within-subject across all samples, and then averaged across subjects. In empirical data, probability densities were estimated across all subjects due to sparse within-subject RT counts.

*e.g., https://git.mpib-berlin.mpg.de/LNDG/stateswitch/behavior/hddm/-/blob/Paper1/A_scripts/B1_HDDM_modeling_EEG_YA_load.ipynb

4. The gamma frequency band seems to be defined differently for different analyses (i.e. 40-90 Hz in Fig. 3; 60-150 Hz in Fig. 4), and indeed the effects in Fig. 4 seem to lie primarily outside of the band focussed on in Fig. 3. Ideally these would be consistent. Is there a reason why the Fig. 3 analysis did not extend higher than 90 Hz. Indeed, the brainscores seem to be highest toward the upper bound of the examined range (Fig. 3D), suggesting some meaningful signal may lie above the upper bound of that reported and may indeed reconcile the results in Figs. 3 and 4.

We thank the reviewer for highlighting this inconsistency. The reviewer correctly assumes that there was no rigorous argument for not including power values above 90 Hz in the PLS analysis (Figure 3). Indeed, an extension of the gamma range supports the reviewer’s observation that the load effect may be even stronger past 90 Hz (Figure R4). In general, the results suggest that gamma sensitivity is indeed broadband-like, as opposed to a narrowband “rhythmic” gamma response.

Figure R4. PLS with extended gamma band up to 140 Hz. See Figure 3D for the corresponding caption.

Notably, extending the gamma range up to 140 Hz did not substantially change the individual brainscores, as we observed high consistencies with the individual brainscores obtained from the more constrained gamma solution (avg. across conditions: $R^2 = .87$; linear effect of target load: $R^2 = .80$). Given the high interindividual reliability with the original brainscore, we did not re-run analyses with this broader definition and report this result only in this reviewer letter. We hope that the slight deviation in gamma frequency ranges that were included across analyses is acceptable and further note that the derivation of gamma-band signals differs in any case between the PLS and cross-frequency coupling analyses.

5. The distinction between the ‘task PLS’ and the ‘behavioural PLS’ is not very clear in the main text – in particular, it is introduced early on with passing reference to supplementary materials (line 199), but then becomes quite important in light of the various findings implicating specifically thalamic BOLD responses in adjustments to task load (Fig. 7). Indeed, I find even the additional explanation given in text S4 to be quite unclear and it makes reference to supplementary figure panels (Fig. S4A/B) that I do not know how to interpret given the scant information in the associated legend. In sum then, some additional clarification in the main text about what differentiates task PLS from behavioural PLS and how each should be interpreted is required, early on where they are first introduced.

We have added a short note regarding the difference between task and behavioral PLS approaches where they first appear for the EEG analyses: “The main difference between task and behavioral PLS rests in the relation of multivariate neural values to categorical design variables in the former, and continuous individual ‘behavioral’ variables in the latter (Krishnan, Williams, McIntosh, & Abdi, 2011).” (l.295ff.). Specifically, the main difference relates to the multivariate target relation to (a) categorical condition labels (for Task PLS), and (b) continuous behavioral values (for Behavioural PLS), respectively (see Krishnan et al., 2011 for an accessible methods review). We have also extended the associated Supplemental Text (now S5) to make this distinction more apparent.

We apologize that the Figure S4A caption was incorrect; it should have noted ‘task PLS’ instead of ‘behavioural PLS’. This may have added to the confusion regarding the distinction between task and behavioral PLS that has been noted by the reviewer. We have extended the previously sparse figure caption to clarify what is shown.

Minor issues:

-- Intro, line 35: "parallel" here seems misleading since EEG and fMRI were measured in separate sessions

We have clarified this statement and removed the term "parallel": "We performed a multi-modal EEG-fMRI experiment, measuring the same participants in two separate sessions, to capture both fast cortical dynamics (EEG) and subcortical activity (fMRI) ..." (l. 35, ff.).

-- Intro, line 47: "broad sensitivity increases". What is meant by this term? Sensitivity in the decision-making literature is a loaded term, typically referring to an increase in perceptual sensitivity captured by measures like d' . Perhaps consider rephrasing.

d' indeed does not capture our intended use of the term sensitivity here, as d' rather captures the signal-to-noise ratio (SNR) in signal detection akin to the gain control mechanism raised in the introduction. Our previous use of "sensitivity" was centered on increased activation for multiple features as illustrated in Fig. 1A. To help clarify in the revision, we have thus changed the phrasing from 'broad sensitivity increases' to the more ambiguous "joint encoding of multiple features", which leaves open the question whether there is (a) a general boost in noise (benefitting multiple stimuli), or (b) a joint increase in multiple feature-specific signals in the presence of stable noise. While the notion of altered E/I balance may suggest the former (thus contradicting the specific notion of d' as an increase in SNR), dissociating these two explanations appears beyond our present paradigm and design.

-- Line 101: typically the acronym "CPP" refers to centro-parietal positivity, not centroparietal positive potential (cf. the O'Connell, Kelly papers where this component was first described).

Our apologies, we have now replaced the term "Centroparietal Positive Potential" with "Centro-Parietal Positivity" throughout the manuscript.

-- Line 147: please define "response agreement". I think the term "response convergence" is also used at various other parts of the manuscript – please pick one and use consistently throughout.

We now use "response convergence" throughout and added a short definition: "trials in which the correct choices (i.e., prevalent options) of all cued features converges on the same button response". We have also expanded on the definition in Text S3: "To reduce response mapping demands following probe presentation, we fixed response mapping for the two options of each feature throughout the experiment. Notably, the correct choices (i.e., prevalent options) for multiple cued features can converge on the same left or right response in a given trial ("response convergence")."

-- Fig S2A: is y-axis of inset meant to be CPP slope, or μ/β slope? Also, legend for panel A refers to subpanels A/B which I think don't exist.

This was indeed an error and we thank the reviewer for spotting it. The y-axis should read " μ/β -slope", which we have now corrected. In the legend, we have now relabeled the insets according to their location in the panel.

-- Fig S3: there seems to be some confusion of panel labels between the figure and the figure legend. Also, there is not a consistent description of what the arrows reflect in panels B and C.

Thank you for indicating problems with the legend. Upon revisiting the legend, we agree that the labels and description were suboptimal. We revised the description in the caption such that the illustrated content can now hopefully be better understood also without reference to the associated text.

-- The phrase "selective attention" or "selective attention condition" is used at several points in the main text (e.g. p.9) without explicit description of what this actually means. I assume it is meant to refer to the single-target condition, where participants can selectively attend to a single feature. But this is not clear, and also I think not that helpful (one could argue that all four levels of load vary in the 'degree' of selective attention that they permit, so I don't think it makes much sense to designate only one level as the 'selective attention condition'). I suggest sticking closer to the actual design and simply refer to this as the single-target condition; further clarification about the high level of selective attention that this condition, relative to the others, permits can then be delivered as desired.

Thank you for this comment. We agree with the rationale that all conditions may require some degree of selectivity; and have implemented the more precise phrasing of the reviewer to better align with the actual design.

-- Fig. 3A: references to "theta" should rather be "delta/theta", since the frequency band in question (2-8 Hz) covers both delta (usually ~1-3 Hz) and theta (usually ~4-8 Hz) bands.

We agree with the reviewer that the common subdivision into clearly separable and fixed frequency bands may be somewhat illusory. We opted to include an extended frequency range at the lower end as it provides the PLS analysis the potential to detect relations, or to indicate that no relation exists (as well as to account for the wavelet-induced frequency spread). While this approach allows sensitivity also to fluctuations traditionally referred to as falling within the delta frequency range, we would like to note that multiple results suggest an effect specificity in the theta range. First, the mediofrontal location of the effect is in line with the literature on mediofrontal theta in studies on EEG cognitive control (e.g., see review by Cavanagh & Frank, 2014). Inspecting the FFT-derived power spectrum at those sensors, we indeed observe a narrowband peak (suggesting the partial presence of rhythms) in the theta, but not the delta frequency range (Figure R5A). Moreover, sensitivity to probe uncertainty as derived from the task PLS was maximal at theta, not delta frequencies (Figure R5B), despite the intrinsic spectral smoothing by the wavelet. Finally, we observed an effect also for the rhythm-specific theta estimates (see Figure S4E), that by definition attempt to exclude the delta band. We argue that these results suggest an effect sensitivity to theta rhythms and believe that a 'delta' label could rather be misleading here. However, should the reviewer feel strongly about this point, we would also be willing to change the terminology to the less specific "delta/theta" descriptor.

Figure R5. Low-frequency rhythm effects are predominantly localized to the theta (~4-8 Hz), not delta (~1-3 Hz) frequency range. (A) Spectral power in low frequency range at mediofrontal channels (shown in the bottom of Figure 3A) indicates a narrowband peak in the theta, but not the delta frequency range. Shaded bars indicate standard errors of the mean. (B) At the same frontal channels, task PLS brainscores (based on the multivariate PLS) are maximal at theta frequencies, suggesting highest sensitivity to probe uncertainty in the theta range.

-- Fig. S4: there's a lot going on in this figure and the legend is extremely brief. The legend entries for several individual panels (in particular A-C) need considerable elaboration.

We have extended the formerly very brief legend captions for panels A-C (replicated below).

Figure S4A-C. Additional spectral power analyses prior and during stimulus presentation. (A) Task PLS results describing multivariate spectral power changes from the pre-stimulus baseline ('central fixation'). This task PLS model targets optimal multivariate spectral power differences between stimulus presentation and pre-stimulus baseline periods. Upper panel: Bootstrap ratios (BSRs) averaged across the respective channel clusters indicated in Figure 3A. Lower panel: individual brainscore changes as a function of stimulus onset. The model indicates joint theta/gamma power increases (positive loadings in red) and alpha power decreases (negative loadings in blue) following stimulus onset as compared to pre-stimulus baseline. BSR = bootstrap ratio (see methods). (B) Behavioral PLS results, linking linear multivariate spectral power changes with target # to drift

rate decreases and pupil diameter modulation. This behavioral PLS model targets the optimal statistical relation between multivariate spectral power data (specifically, linear changes as a function of target load) and individual differences in drift rate and pupil modulations. Upper panel: Bootstrap ratios (BSRs) averaged across the respective channel clusters indicated in Figure 3A. Lower panel: Individual brainscore relations to linear changes in drift rate and pupil diameter. **(C) Parieto-occipital pre-stimulus alpha power increases with target load but is not related to drift changes (see Text S4).** The time-frequency plot shows the results of the task PLS relating multivariate pre-stimulus ('fixation') power values to target load categories, indicating major loadings in the alpha band. The inset shows the topography of mean BSR values and the channels across which results were averaged for the time-frequency plot. The lower plot illustrates brainscore increases as a function of target number. Linear brainscore increases were not related to drift changes (not shown here, see Text S4).

-- Lines 242-244: "...the magnitude of individual entropy modulation in this cluster scaled with increases in the SPMF, indicating that alpha desynchronization was accompanied by broadband changes in signal irregularity". From what I understand the SPMF is a composite multivariate measure that incorporates relationships between frequency-specific power changes and target load (uncertainty) across *all* considered frequencies. So it is non-specific in the sense that it does not parse theta-band from alpha-band contributions. How, then, can an entropy/SPMF correlation specifically support the inference of an alpha/entropy relationship?

The reviewer correctly criticizes the wording or the implied direct relationship. Alpha is not exclusively part of this multivariate score, and as such the results are not specific to alpha. However, the intended inference –that there is a relationship of alpha to entropy as part of the multivariate score– still holds, though this relationship is not an exclusive one. We have clarified the wording accordingly: "This indicates that multivariate changes in spectral power, including alpha desynchronization, were accompanied by broadband changes in signal irregularity."

-- Line 296: "fast integrators" is not very clear. I suggest replacing with "individuals with higher drift rates". After all, drift rate captures more than just integration speed – it's really a measure of the relative signal-to-noise (assuming a fixed within-trial noise parameter) of the evidence source.

We agree with this rationale and have changed the wording accordingly.

-- Line 299: "This suggests that arousal...related to increases in local cortical excitability..." This seems too strong given that the only observed relationship is between pupil and SPMF, and *not* with putatively more specific measures of excitability (entropy and power spectrum slope). On a related note, was there any frequency-specificity to the pupil effect, or did pupil correlate with the range of spectral modulations that combine to make up the SPMF effect?

We changed the wording to the more specific 'spectral power changes' given that we observed no direct relation to entropy or aperiodic slope modulation. Notably, we observed a joint relation with entropy modulation in the final multi-modal model, suggesting at least some shared variance amongst this triad.

The pupil response correlated with the entire range of spectral modulations that combined make up the SPMF effect. We confirmed this using a behavioral PLS (see Figure S4B), but this may have suffered from the presentation issue noted above. Note that the behavioral PLS does not indicate that the strength of association was equal across frequencies, only that there was shared individual variation across frequencies that related to the pupil effect. To specifically assess the relation of pupil diameter modulation, we ran another more selective behavioral PLS that included three different variants of pupil diameter assessment: (a) the first derivative used throughout the main manuscript, (b) the 'raw' pupil diameter and (c) a variant

that has been baseline-corrected. Results are shown in Figure R6 and indicate a significant LV that dominantly loaded on frontal delta/theta and posterior gamma, with less of a link to posterior alpha, while also highlighting high convergence between different definitions of the pupil effect. Especially around the time of the pupil diameter effect (0.5 to 1 sec post-stim), the relation appears constrained to the delta/theta range.

Figure R6. Behavioral PLS relating linear changes in spectral power as a function of target load to individual differences in linear increases in pupil diameter measures. (A) Mean BSR values, thresholded at a BSR of 3 (see caption of Figure 3D). Topographies illustrate the channels selected for plotting (same as in the main analysis). (B) Linear relations between linear pupil modulations as a function of target load and brainscores (reflecting underlying linear target load modulation of spectral power).

-- Lines 649-650: RGB values seem to be incorrect. Specifically, the value for 'low saturation green' will produce a red colour, while 'high saturation red' will produce a green colour.

While the values were correct, the associated labels were indeed wrong. We thank the reviewer for spotting this error and have corrected the text accordingly.

-- P. 31-32: why was baselining applied to gamma-band estimates but not lower frequencies?

We very much appreciate this question and hope that the following more detailed response provides clarification. As the reviewer can likely appreciate, temporal baselining can be rather problematic, as it introduces relationships between different time periods. Especially given the pre-stimulus alpha effect, this could lead to unintuitive reflections during the stimulus presentation period. For the low-frequency ranges, we thus exclusively use single-trial log10 transformations, in line with previous recommendations (Smulders, ten Oever, Donkers, Quaedflieg, & van de Ven, 2018).

The gamma-band range is well-known for its' sensitivity to non-neural sources (e.g., Hipp & Siegel, 2013). Our baselining procedure aimed to extract the neural signal component, under the assumption that non-neural sources should dominate gamma power variation in the absence of visual stimulation (i.e., during pre-stimulus fixation). We therefore consider this 'gamma' power measure a 'gamma-band' SNR score, although we agree it should be validated more thoroughly in data with known visual gamma-band responses to truly interpret it as such. In particular, the mean during stimulus presentation is normalized by the amount of variation across trials pre-stimulus. This assumes that e.g., muscle contributions would show strong variability across trials. Indeed, this normalization suggested a clear posterior increase in gamma power, which was less apparent without normalization (Figure R7). We perceive the presence of gamma power increases during stimulus presentation as a prerequisite for the subsequent assessment of a target uncertainty effect, which is why we opted for the

baseline normalization approach for subsequent gamma-band analyses. We now briefly note the above in the methods section.

Figure R7. Comparison of “raw” (A) and baseline-normalized (B) gamma power during stimulus processing. (Top) Topography of average gamma power (> 40 Hz) during stimulus presentation (shown in green below). Posterior gamma power is clearly visible only for normalized estimates. (Bottom) Time-frequency representation of log₁₀-transformed (A) and normalized (B) spectral power. Power values are averaged across posterior channels indicated on top. Increased broadband gamma power at posterior channels is clearly visible following SNR baseline normalization in B, but not for raw estimates.

-- Line 817: reference to Figure 1A seems to be incorrect.

This was indeed missing the corresponding paper reference (Kosciessa et al., 2020) in which the Figure appears. We apologize for this oversight and have now added the reference.

Reviewer #3:

Thank you for inviting me to review this manuscript by Kosciessa and colleagues, in which the authors conducted a series of computational neuroimaging studies in order to better understand how the brain deals with uncertainty.

The authors hypothesized that uncertainty would facilitate a shift from selective to diffuse, asynchronous processing, which they further suggest should align with activity in the thalamus. To test these hypotheses, the authors had participants perform a parametric adaptation of the classic dot motion task, in which they manipulated the number of stimulus dimensions that were task-relevant in a given trial while holding the sensory features of the task (i.e., its appearance on the screen) constant across trials. Probe uncertainty was parametrically manipulated using valid pre-stimulus cues, indicating the feature set from which a probe would be selected.

The authors then use an impressive range of different computational and statistical approaches to conclude that task uncertainty is related to lowered drift rate, asynchronous EEG signatures, dilated pupils and upregulated activity in the thalamus. They conclude that neuromodulatory processes involving the thalamus shape cortical excitability states in humans, and that a shift from alpha-rhythmic to aperiodic neural dynamics adjusts the processing fidelity of external stimuli in service of upcoming decisions.

Overall, I found the manuscript to be clear, well-motivated and professionally executed. I have only minor questions for the authors that I hope will help to clarify the main results.

We thank the reviewer for the review. We hope that our revision and responses below successfully clarify the open issues.

1. I didn't quite understand the rationale for fixing the threshold across target load levels in the model. Surely the threshold was found to change loads? If so, is it not an important feature of any resultant model? Do the authors results hold if the measure is not held constant?

In the model that incorporated threshold variations, the threshold indeed was found to vary (see Supplementary Figure S1C). However, despite this model obtaining a better DIC fit, we fixed the threshold parameter across conditions based on the EEG data (for a similar approach, see McGovern, Hayes, Kelly, & O'Connell, 2018). The rationale here is that the level to which evidence is integrated should serve as a proxy for threshold (or "boundary separation") differences. If the boundary separation is larger, more evidence should be required to commit to a decision, and as such, the threshold (and level to which the CPP integrates) should be larger. However, we did not observe differences in the CPP level at response, nor did we see differences in contralateral motor preparation (mu-beta desynchronization) level. Assuming such relations (e.g., McGovern et al., 2018) this provides no electrophysiological evidence for uncertainty-related changes in boundary separation (albeit absence of evidence is not evidence of absence).

Notably, the assumption that CPP or mu-beta level at response capture differences in evidence integration threshold or boundary separation may be wrong or sub-optimally assessed in the present study. This may best be tested in a parameter recovery study, rather than in a novel paradigm as we used here. Crucially, regardless of whether the threshold is chosen to vary freely or not (the latter being our choice in the current manuscript), the main results concerning drift rate and its modulation as a function of target load are highly stable. That is, both the average group effect, as well as individual variation in the magnitude of linear drift rate adaptations (average drift rate across target load: EEG: $R^2 = .98$, MRI: $R^2 = .98$; linear drift rate change: EEG: $R^2 = .90$; MRI: $R^2 = .86$) remain highly similar independent of the

ultimate model choice between in- or excluding threshold variations as a function of target load. In sum, while the effect of uncertainty on thresholds (and ultimately response bias) and the association between EEG and DDM parameters remains somewhat unclear, this does not directly affect our statements regarding the drift rate effect of interest that is at the center of the current study. This also motivated the inclusion and discussion of these details in the Supplement.

2. Drift rate looks like it shows a non-linear effect of parametric uncertainty, whereas NDT smoothly increased as a function of load. Do the authors have an explanation for why the two measures showed different interactions with parametric uncertainty?

Based on our supplementary analysis summarized in Figure S3, we suggest that the NDT effect largely reflects a parametric increase in late-stage preparation of the motor execution, perhaps partly related to the fixed button mapping (e.g., red = left, green = right) across the experiment that we used to avoid additional remapping demands. Our results hint at the possibility that this worked well when only a single target was relevant, while such “remapping demands” may have persisted for multiple relevant targets. As a prediction of this observation, we would expect similar NDT increases also for the single-target condition, if the button mapping was randomly assigned during each trial, rather than fixed across the entire experiment. As a side note, we observed specific correlations between NDT increases and RTs, as well as drift rate decreases and accuracy reductions (see Text S1), hinting at some dissociation also at the level of ‘raw’ behavioral indicators.

In contrast, we suggest that drift rate represents the non-linear uncertainty effect of interest, with the strongest modulation observed when moving from certainty to uncertainty. This is in line with similar non-linearities observed in the neural target signatures of interest, albeit with some noted variation in whether post-hoc statistics indicated residual differences between higher target load levels. We consider the non-linearities sensible for multiple reasons: (a) Given that the task becomes increasingly demanding, at least some subjects may show decreasing task engagement. This is evidenced by the observed decrease in frontoparietal BOLD signal at higher target loads (LV2 in task PLS; Figure S5A2), which in turn may produce somewhat diminishing returns in the processes that boost excitability. (b) The psychometric properties of individual features were not individually calibrated. As such, we cannot rule out that some features may have been harder to discriminate than others, which may have affected their joint assessment especially towards higher target load levels (please also see our response to Reviewer 1’s Q3 in which we address this issue in greater detail).

3. I'm sorry if I missed this, but how stable were the results from the hDDM model? What kind of model search did the authors use so as to ensure stability?

Given that there are multiple notions of model stability, we provide a response to two different interpretations below.

The first interpretation concerns the notion of model comparison as raised in question 1 above. We assessed the Deviance Information Criterion (DIC) index for models that were independently run for EEG and MRI sessions. Results from this model comparison are shown in Figure S1B. As described above, we selected the model that was in accordance with our EEG results, which argued against differences in threshold as a function of target load. However, our main (e.g., multi-modal) results remained highly stable when choosing a different model (see response to question 1). Running the models separately within each session allowed us to assess the stability/reliability of individual estimates for the chosen model (see Figure S1F).

There is another interpretation of ‘stability’ when it comes to model evaluation, namely stability to starting point of the search algorithm. HDDM intrinsically uses Markov-Chain Monte Carlo samples to estimate parameters, and as noted in the manuscript, we estimated 7000

samples, discarding a liberal amount of 5000 initial samples to achieve convergence (and thus an independence from the starting point parameter constellation). We visually inspected the samples to confirm the absence of slow drifts or large variation. This was done for each of the reported models. For the interested reader, MCMC traces are also available for post-hoc inspection via the shared Jupyter notebooks (e.g., https://git.mpib-berlin.mpg.de/LNDG/switch/behavior/hddm/-/blob/Paper1/A_scripts/B1_HDDM_modeling_EEG_YA_load.ipynb).

Related to confidence in model stability, we additionally observed relations between inter-individual differences in drift rate and CPP slopes.

We have now added qualitative depictions of the model fit also in response to Reviewer 2's Q3 (see Figure S1G).

4. The thalamic loadings in the first latent component of the PLS in Figure 7D1 are reminiscent of Ted Jones' "matrix" regions (Jones, 2001). The authors may wish to comment on this potential overlap, which can be quantified using the calcium binding protein Calbindin (Niemann et al., 2000; Müller et al., 2020).

Various classification schemes exist for intra-thalamic circuit differences (Halassa & Kastner, 2017; Halassa & Sherman, 2019), with the "matrix" vs. "core" cell distinction (Jones, 2001, 1998) being one of the most prominent. With regard to our present work, we consider their distinction to be most interesting with regard to their relative importance for relay and modulation, respectively. Whereas "core" cells feed sensory-motor information forward to granular cortical layers in a regionally-specific fashion, "matrix" cells can diffusely target superficial cortical layers (Cruikshank et al., 2012; Jones, 1998) and may align cortical membrane excitability to momentary behavioural goals (Rikhye, Wimmer, & Halassa, 2018) without necessarily eliciting activity (Reichova & Sherman, 2004). This distinction complements a classification scheme according to different nuclei (Halassa & Sherman, 2019; Sherman & Guillery, 2013); whereas relay nuclei respond maximally to transients, such as the on- and offsets of sensory stimuli (Alonso & Swadlow, 2005; Bruno & Sakmann, 2006; Rose & Metherate, 2005; Theyel, Llano, & Sherman, 2010), modulatory (or 'higher-order') nuclei increase activity particularly during high cognitive demands (Bolkan et al., 2018; Cruikshank et al., 2012; Delevich, Tucciarone, Huang, & Li, 2015; Parnaudeau et al., 2013; Schmitt et al., 2017). Notably, as we now discuss in the main text, we observed maximal loadings in higher-order nuclei that are thought to modulate cortical connectivity (Halassa & Kastner, 2017) and which we suggest are crucial for the top-down attention modulation observed in our findings.

However, as the reviewer notes, this begs the question whether a "matrix" vs. "core" classification may similarly or better capture the main effect of probe uncertainty and the behavioral relations observed in our study. We explored this question further using the quantification of Calbindin-to-Parvalbumin kindly provided by E. J. Muller et al. (2020). Specifically, we explored whether uncertainty-related increases from the task PLS and results from the behavioral PLS mapped onto a connectivity gradient of Matrix-to-Core cells in thalamus. We used a published map of CP_T that provides a proxy measure of the relative amount of Calbindin (CALB1) to Parvalbumin (PVALB) expression based on the Allen Brain Atlas, and therefore a proxy of the relative amount of thalamic Matrix-to-Core projection cells in the MRI volume (E. J. Muller et al., 2020). We resliced the available map to 3mm to align with our bootstrap ratio scores. [We also noted a predominance of negative values in the provided map, contrasting with Figure 1B of E. J. Muller et al. (2020). For this reason, we rescaled values to the range [-1, 1]. This only affects the absolute value ranges, not the derivation of the relative scores however.] Subsequently, we correlated CP_T values with bootstrap ratios across all voxels within the thalamic volume. As shown in Figure R8, loadings were consistently highest for mixture populations of calbindin and parvalbumin in both the task and especially the behavioral PLS, but surprisingly not for voxels predominantly expressing Calbindin (suggesting the prevalence of Matrix cells). Results were qualitatively similar without

prior rescaling. This pattern of results makes a substantive conclusion regarding the relevance of “matrix” and “core” cells in our results difficult, and we thus report this analysis only in this reviewer letter. Specifically, these results speak against a clear conclusion that thalamic loadings are predominant in “matrix”-dominant regions of the thalamus, while suggesting a potentially more complex relation to the regional mixture of Calbindin- and Parvalbumin-expressing cells.

Figure R8. Relation of thalamic bootstrap ratios and CP_{τ} (relative calbindin-to-parvalbumin) scores. Only negative CP_{τ} values show a consistent relation with bootstrap ratios from the PLS across both the task (A) and behavioral PLS (B).

References

- Alonso, J. M., & Swadlow, H. A. (2005). Thalamocortical specificity and the synthesis of sensory cortical receptive fields. *Journal of Neurophysiology*, *94*(1), 26-32. doi:10.1152/jn.01281.2004
- Atallah, B. V., & Scanziani, M. (2009). Instantaneous Modulation of Gamma Oscillation Frequency by Balancing Excitation with Inhibition. *Neuron*, *62*(4), 566-577. doi:10.1016/j.neuron.2009.04.027
- Bolkan, S. S., Stujenske, J. M., Parnaudeau, S., Spellman, T. J., Rauffenbart, C., Abbas, A. I., . . . Kellendonk, C. (2018). Thalamic projections sustain prefrontal activity during working memory maintenance (vol 20, pg 987, 2017). *Nature Neuroscience*, *21*(8), 1138-1138. doi:10.1038/s41593-018-0132-2
- Bruno, R. M., & Sakmann, B. (2006). Cortex is driven by weak but synchronously active thalamocortical synapses. *Science*, *312*(5780), 1622-1627. doi:10.1126/science.1124593
- Cavanagh, J. F., & Frank, M. J. (2014). Frontal theta as a mechanism for cognitive control. *Trends in Cognitive Sciences*, *18*(8), 414-421. doi:10.1016/j.tics.2014.04.012
- Chakraborty, S., Kolling, N., Walton, M. E., & Mitchell, A. S. (2016). Critical role for the mediodorsal thalamus in permitting rapid reward-guided updating in stochastic reward environments. *Elife*, *5*. doi:10.7554/eLife.13588
- Cruikshank, S. J., Ahmed, O. J., Stevens, T. R., Patrick, S. L., Gonzalez, A. N., Elmaleh, M., & Connors, B. W. (2012). Thalamic Control of Layer 1 Circuits in Prefrontal Cortex. *Journal of Neuroscience*, *32*(49), 17813-17823. doi:10.1523/Jneurosci.3231-12.2012
- Delevich, K., Tucciarone, J., Huang, Z. J., & Li, B. (2015). The Mediodorsal Thalamus Drives Feedforward Inhibition in the Anterior Cingulate Cortex via Parvalbumin Interneurons. *Journal of Neuroscience*, *35*(14), 5743-5753. doi:10.1523/Jneurosci.4565-14.2015
- Destexhe, A., Rudolph, M., & Pare, D. (2003). The high-conductance state of neocortical neurons in vivo. *Nature Reviews: Neuroscience*, *4*(9), 739-751. doi:10.1038/nrn1198
- Edelstyn, N. M. J., Mayes, A. R., & Ellis, S. J. (2014). Damage to the dorsomedial thalamic nucleus, central lateral intralaminar thalamic nucleus, and midline thalamic nuclei on the right-side impair executive function and attention under conditions of high demand but not low demand. *Neurocase*, *20*(2), 121-132. doi:10.1080/13554794.2012.713497
- Fries, P. (2015). Rhythms for Cognition: Communication through Coherence. *Neuron*, *88*(1), 220-235. doi:10.1016/j.neuron.2015.09.034
- Gao, R., Peterson, E. J., & Voytek, B. (2017). Inferring synaptic excitation/inhibition balance from field potentials. *NeuroImage*, *158*, 70-78. doi:10.1016/j.neuroimage.2017.06.078
- Gold, J. I., & Shadlen, M. N. (2007). The neural basis of decision making. *Annual Review of Neuroscience*, *30*, 535-574. doi:10.1146/annurev.neuro.29.051605.113038
- Haegens, S., Nacher, V., Luna, R., Romo, R., & Jensen, O. (2011). alpha-Oscillations in the monkey sensorimotor network influence discrimination performance by rhythmical inhibition of neuronal spiking. *Proceedings of the National Academy of Sciences of the United States of America*, *108*(48), 19377-19382. doi:10.1073/pnas.1117190108
- Halassa, M. M., & Kastner, S. (2017). Thalamic functions in distributed cognitive control. *Nature Neuroscience*, *20*(12), 1669-1679. doi:10.1038/s41593-017-0020-1
- Halassa, M. M., & Sherman, S. M. (2019). Thalamocortical Circuit Motifs: A General Framework. *Neuron*, *103*(5), 762-770. doi:10.1016/j.neuron.2019.06.005
- Hipp, J. F., & Siegel, M. (2013). Dissociating neuronal gamma-band activity from cranial and ocular muscle activity in EEG. *Frontiers in Human Neuroscience*, *7*. doi:10.3389/fnhum.2013.00338
- Jaramillo, J., Mejias, J. F., & Wang, X. J. (2019). Engagement of Pulvino-cortical Feedforward and Feedback Pathways in Cognitive Computations. *Neuron*, *101*(2), 321-336 e329. doi:10.1016/j.neuron.2018.11.023
- Jones, E. G. (1998). Viewpoint: the core and matrix of thalamic organization. *Neuroscience*, *85*(2), 331-345. doi:10.1016/s0306-4522(97)00581-2
- Jones, E. G. (2001). The thalamic matrix and thalamocortical synchrony. *Trends in Neurosciences*, *24*(10), 595-601. doi:10.1016/S0166-2236(00)01922-6
- Klimesch, W., Sauseng, P., & Hanslmayr, S. (2007). EEG alpha oscillations: the inhibition-timing hypothesis. *Brain Research Reviews*, *53*(1), 63-88. doi:10.1016/j.brainresrev.2006.06.003
- Komura, Y., Nikkuni, A., Hirashima, N., Uetake, T., & Miyamoto, A. (2013). Responses of pulvinar neurons reflect a subject's confidence in visual categorization. *Nature Neuroscience*, *16*(6), 749-755. doi:10.1038/nn.3393
- Kosciessa, J. Q., Kloosterman, N. A., & Garrett, D. D. (2020). Standard multiscale entropy reflects neural dynamics at mismatched temporal scales: What's signal irregularity got to do with it? *PLoS Computational Biology*, *16*(5), e1007885. doi:10.1371/journal.pcbi.1007885

- Krishnan, A., Williams, L. J., McIntosh, A. R., & Abdi, H. (2011). Partial Least Squares (PLS) methods for neuroimaging: a tutorial and review. *NeuroImage*, *56*(2), 455-475. doi:10.1016/j.neuroimage.2010.07.034
- Lorincz, M. L., Kekesi, K. A., Juhasz, G., Crunelli, V., & Hughes, S. W. (2009). Temporal Framing of Thalamic Relay-Mode Firing by Phasic Inhibition during the Alpha Rhythm. *Neuron*, *63*(5), 683-696. doi:10.1016/j.neuron.2009.08.012
- Mack, M. L., Preston, A. R., & Love, B. C. (2020). Ventromedial prefrontal cortex compression during concept learning. *Nat Commun*, *11*(1), 46. doi:10.1038/s41467-019-13930-8
- Marton, T. F., Seifkar, H., Luongo, F. J., Lee, A. T., & Sohal, V. S. (2018). Roles of Prefrontal Cortex and Mediodorsal Thalamus in Task Engagement and Behavioral Flexibility. *Journal of Neuroscience*, *38*(10), 2569-2578. doi:10.1523/JNEUROSCI.1728-17.2018
- McGovern, D. P., Hayes, A., Kelly, S. P., & O'Connell, R. G. (2018). Reconciling age-related changes in behavioural and neural indices of human perceptual decision-making. *Nat Hum Behav*, *2*(12), 955-966. doi:10.1038/s41562-018-0465-6
- Mitchell, A. S. (2015). The mediodorsal thalamus as a higher order thalamic relay nucleus important for learning and decision-making. *Neuroscience and Biobehavioral Reviews*, *54*, 76-88. doi:10.1016/j.neubiorev.2015.03.001
- Mo, C., Lu, J., Wu, B., Jia, J., Luo, H., & Fang, F. (2019). Competing rhythmic neural representations of orientations during concurrent attention to multiple orientation features. *Nat Commun*, *10*(1), 5264. doi:10.1038/s41467-019-13282-3
- Muller, E. J., Munn, B., Hearne, L. J., Smith, J. B., Fulcher, B., Arnatkeviciute, A., . . . Shine, J. M. (2020). Core and matrix thalamic sub-populations relate to spatio-temporal cortical connectivity gradients. *NeuroImage*, *222*, 117224. doi:10.1016/j.neuroimage.2020.117224
- Muller, K. R., Mika, S., Ratsch, G., Tsuda, K., & Scholkopf, B. (2001). An introduction to kernel-based learning algorithms. *IEEE Transactions on Neural Networks*, *12*(2), 181-201. doi:10.1109/72.914517
- Parker, A., Eacott, M. J., & Gaffan, D. (1997). The recognition memory deficit caused by mediodorsal thalamic lesion in non-human primates: A comparison with rhinal cortex lesion. *European Journal of Neuroscience*, *9*(11), 2423-2431. doi:10.1111/j.1460-9568.1997.tb01659.x
- Parnaudeau, S., O'Neill, P. K., Bolkan, S. S., Ward, R. D., Abbas, A. I., Roth, B. L., . . . Kellendonk, C. (2013). Inhibition of mediodorsal thalamus disrupts thalamofrontal connectivity and cognition. *Neuron*, *77*(6), 1151-1162. doi:10.1016/j.neuron.2013.01.038
- Pergola, G., Danet, L., Pitel, A. L., Carlesimo, G. A., Segobin, S., Pariente, J., . . . Barbeau, E. J. (2018). The Regulatory Role of the Human Mediodorsal Thalamus. *Trends in Cognitive Sciences*, *22*(11), 1011-1025. doi:10.1016/j.tics.2018.08.006
- Pettine, W. W., Louie, K., Murray, J. D., & Wang, X.-J. (2020). Hierarchical Network Model Excitatory-Inhibitory Tone Shapes Alternative Strategies for Different Degrees of Uncertainty in Multi-Attribute Decisions. *bioRxiv*.
- Podvalny, E., Noy, N., Harel, M., Bickel, S., Chechik, G., Schroeder, C. E., . . . Malach, R. (2015). A unifying principle underlying the extracellular field potential spectral responses in the human cortex. *Journal of Neurophysiology*, *114*(1), 505-519. doi:10.1152/jn.00943.2014
- Poo, C., & Isaacson, J. S. (2009). Odor representations in olfactory cortex: "sparse" coding, global inhibition, and oscillations. *Neuron*, *62*(6), 850-861. doi:10.1016/j.neuron.2009.05.022
- Reichova, I., & Sherman, S. M. (2004). Somatosensory corticothalamic projections: Distinguishing drivers from modulators. *Journal of Neurophysiology*, *92*(4), 2185-2197. doi:10.1152/jn.00322.2004
- Reimer, J., Froudarakis, E., Cadwell, C. R., Yatsenko, D., Denfield, G. H., & Tolias, A. S. (2014). Pupil fluctuations track fast switching of cortical states during quiet wakefulness. *Neuron*, *84*(2), 355-362. doi:10.1016/j.neuron.2014.09.033
- Rigotti, M., Barak, O., Warden, M. R., Wang, X. J., Daw, N. D., Miller, E. K., & Fusi, S. (2013). The importance of mixed selectivity in complex cognitive tasks. *Nature*, *497*(7451), 585-590. doi:10.1038/nature12160
- Rikhye, R. V., Gilra, A., & Halassa, M. M. (2018). Thalamic regulation of switching between cortical representations enables cognitive flexibility. *Nature Neuroscience*, *21*(12), 1753-1763. doi:10.1038/s41593-018-0269-z
- Rikhye, R. V., Wimmer, R. D., & Halassa, M. M. (2018). Toward an Integrative Theory of Thalamic Function. *Annual Review of Neuroscience*, *Vol 41*, *41*, 163-183. doi:10.1146/annurev-neuro-080317-062144
- Rose, H. J., & Metherate, R. (2005). Auditory thalamocortical transmission is reliable and temporally precise. *Journal of Neurophysiology*, *94*(3), 2019-2030. doi:10.1152/jn.00860.2004

- Saalmann, Y. B., Pinsk, M. A., Wang, L., Li, X., & Kastner, S. (2012). The pulvinar regulates information transmission between cortical areas based on attention demands. *Science*, 337(6095), 753-756. doi:10.1126/science.1223082
- Schmitt, L. I., Wimmer, R. D., Nakajima, M., Happ, M., Mofakham, S., & Halassa, M. M. (2017). Thalamic amplification of cortical connectivity sustains attentional control. *Nature*, 545(7653), 219-223. doi:10.1038/nature22073
- Sherman, S. M., & Guillery, R. W. (2013). Functional Connections of Cortical Areas: A New View from the Thalamus. *Functional Connections of Cortical Areas: A New View from the Thalamus*, 1-279.
- Smulders, F. T. Y., ten Oever, S., Donkers, F. C. L., Quaedflieg, C. W. E. M., & van de Ven, V. (2018). Single-trial log transformation is optimal in frequency analysis of resting EEG alpha. *European Journal of Neuroscience*, 48(7), 2585-2598. doi:10.1111/ejn.13854
- Soltani, A., & Izquierdo, A. (2019). Adaptive learning under expected and unexpected uncertainty. *Nature Reviews Neuroscience*, 20(10), 635-644. doi:10.1038/s41583-019-0180-y
- Spaak, E., Bonnefond, M., Maier, A., Leopold, D. A., & Jensen, O. (2012). Layer-specific entrainment of gamma-band neural activity by the alpha rhythm in monkey visual cortex. *Current Biology*, 22(24), 2313-2318. doi:10.1016/j.cub.2012.10.020
- Theyel, B. B., Llano, D. A., & Sherman, M. (2010). The corticothalamocortical circuit drives higher-order cortex in the mouse. *Nature Neuroscience*, 13(1), 84-U246. doi:10.1038/nn.2449
- Wright, N. F., Vann, S. D., Aggleton, J. P., & Nelson, A. J. (2015). A critical role for the anterior thalamus in directing attention to task-relevant stimuli. *Journal of Neuroscience*, 35(14), 5480-5488. doi:10.1523/JNEUROSCI.4945-14.2015

REVIEWERS' COMMENTS

Reviewer #1 (Remarks to the Author):

The authors have addressed my comments in full and I would like to congratulate them on a terrific study.

Reviewer #2 (Remarks to the Author):

I thank the authors for addressing all of my previous comments, and have no additional concerns.

Reviewer #3 (Remarks to the Author):

The authors should be commended for a thorough revision of their manuscript.

REVIEWERS' COMMENTS

Reviewer #1 (Remarks to the Author):

The authors have addressed my comments in full and I would like to congratulate them on a terrific study.

Reviewer #2 (Remarks to the Author):

I thank the authors for addressing all of my previous comments, and have no additional concerns.

Reviewer #3 (Remarks to the Author):

The authors should be commended for a thorough revision of their manuscript.

We thank the reviewers for their encouraging and constructive comments and are happy that our prior revision satisfactorily addressed all previous comments.